# Bamboo-like dual-phase nanostructured copper composite strengthened by amorphous boron framework

Hang Lv[1,5], Xinxin Gao[1,5], Kan Zhang ◎[1] ✉, Mao Wen ◎[1], Xingjia He[1], Zhongzhen Wu[2], Chang Liu ◎[3] ✉, Changfeng Chen[4] & Weitao Zheng ◎[1]

Grain boundary engineering is a versatile tool for strengthening materials by tuning the composition and bonding structure at the interface of neighboring crystallites, and this method holds special significance for materials composed of small nanograins where the ultimate strength is dominated by grain boundary instead of dislocation motion. Here, we report a large strengthening of a nanocolumnar copper film that comprises columnar nanograins embedded in a bamboo-like boron framework synthesized by magnetron sputtering co-deposition, reaching the high nanoindentation hardness of 10.8 GPa among copper alloys. The boron framework surrounding copper nanograins stabilizes and strengthens the nanocolumnar copper film under indentation, benefiting from the high strength of the amorphous boron framework and the constrained deformation of copper nanocolumns confined by the boron grain boundary. These findings open a new avenue for strengthening metals via construction of dual-phase nanocomposites comprising metal nanograins embedded in a strong and confining light-element grain boundary framework.

The high ductility and malleability make metals indispensable in numerous applications, but these materials, especially most prominent ones like copper and aluminum, are soft and susceptible to tear and wear. Strengthening metals has long been a major topic in materials research with implications across many scientific disciplinaries and technological areas. Traditional methods rely on controlling the generation and interaction of internal defects, such as solute atoms, dislocations, and grain boundaries (GBs)[1–3], among which GBs play an increasing role at decreasing grain sizes down to nanoscale[4]. This phenomenon, known as the Hall-Petch effect[5], has been driving a great deal of research seeking effective ways for strengthening metals. However, when the grain size decreases to a critical value (usually around 10 nm), the proportion of atoms at GBs is high enough to cause a change in plastic deformation from dislocation motion to GB mediated mechanisms, often leading to softening due to structural

weaknesses at the boundary[6]. To strengthen nanocrystalline metals, it is essential to suppress GB induced structural weakness, which has been pursued extensively in recent works[7–11]. Hu et al.[12] successfully stabilized GBs in nanocrystalline nickel-molybdenum alloys by altering the composition of molybdenum and annealing samples at appropriate temperatures, achieving significantly enhanced hardness in samples containing grains of 8.2 nm in size. Li et al.[13] constructed a Schwarz crystal structure in nanocrystalline copper with grain sizes around 10 nm and used the GB relaxation effect to form an interface network between coherent twin boundaries and the large-angle GBs, greatly improving the stability and strength of copper. Wu et al.[14] synthesized nanocrystalline magnesium alloys consisting of nanocrystalline cores of around 6 nm in diameter embedded in amorphous glassy shells that stabilize the GBs and block the propagation of shear bands, producing strengths approaching the ideal theoretical limit.

[1]State Key Laboratory of Superhard Materials, Department of Materials Science and Key Laboratory of Automobile Materials, MOE, Jilin University, Changchun 130012, China. [2]School of Advanced Materials, Peking University Shenzhen Graduate School, Shenzhen 518055, China. [3]International Center for Computational Methods and Software, College of Physics, Jilin University, Changchun 130012, China. [4]Department of Physics and Astronomy, University of Nevada, Las Vegas, NV 89154, USA. [5]These authors contributed equally: Hang Lv, Xinxin Gao. ✉e-mail: kanzhang@jlu.edu.cn; liuchang127@jlu.edu.cn

A notable design strategy for strengthening nanocrystalline metal is to emulate the columnar structures found in nature, such as bamboo[15,16] and honeycomb[17,18], where the highly anisotropic structural arrangements can sustain large mechanical loadings, producing improved strength and hardness. Bamboo has a columnar structure with longitudinally aligned fibers comprising hollow tubular structures[15], generating high strength-to-weight and stiffness-to-weight ratios, as well as high bending strength[16]. Honeycomb structures exhibit similar characteristics with a large number of hollow columnar areas that can host a secondary phase[18], and the cooperation of such a phase and the support of the skeleton structure can sustain high compressive stress applied parallel to the column direction[19]. This strategy holds promise for improving strength and hardness of metals by constructing nanocolumnar films via magnetron sputtering co-deposition[20]. Studies have shown that GB segregation can reduce GB energy, promote grain refinement, and stabilize GBs[21–24]. Meanwhile, phase segregated GBs can be harnessed to serve as the columnar skeleton.

To construct the desired nanocomposite structure, it is key to identify a dual-phase material system that is suitable for both GB segregation and columnar growth. To this end, we have selected copper-boron (Cu-B) system based on the following considerations. First, Cu and B have similar electronegativity and large atomic size mismatch, making it difficult to form an ordered Cu-B alloy or a covalent bonding structure[25,26], which is further diminished by the extremely low solubility of 0.06 at.% of B in Cu at room temperature[27]. Second, Cu alloy thin films tend to have columnar growth mode on Si substrates under appropriate deposition conditions[28]. Third, amorphous boron possesses high strength and hardness[29], providing a favorable basis for forming a strong GB framework.

Here, we report magnetron sputtering co-deposition of a "bamboo-like" dual-phase Cu-B nanocomposite film with nanocolumnar Cu grains embedded in an amorphous boron framework that constitutes an integrated network of GBs. The synthesized film exhibits a very high nanoindentation hardness of 10.8 GPa among all Cu alloys reported to date. Transmission electron microscope (TEM) analysis reveals that the boron GBs impede dislocation motion, and the mechanically strong amorphous boron framework stabilizes columnar copper film containing ultrarefined grains and constrains deformation modes under nanoindentation, switching from the relatively weak shear deformation mode commonly among metals to microstructurally constrained much stronger bending deformation mode, thereby producing improved strength and hardness in the "bamboo-like" nanocomposite Cu-B film. At the same time, this constraint on the shear behavior allows the film to avoid shear-induced failure and ensures that the film has a yield strength of ~1.36 GPa and a flow stress of ~2.58 GPa, as well as a failure strain of over 50%. These results demonstrate an effective approach for strengthening copper via a dual-phase nanostructure design that is expected to be robust and broadly applicable to other metals.

## Results and discussion

### Structural characterization of the dual-phase Cu-B nanocomposite film

We synthesized the "bamboo-like" dual-phase Cu-B nanocomposite film by magnetron sputtering co-deposition and performed detailed structural characterization (see Methods). The atomic concentration of element B in the film is determined to be 26.5 at.% through the utilization of X-ray photoelectron spectroscopy (XPS) analysis (Supplementary Fig. 1). Furthermore, the crystallographic arrangement of the film is identified to be of the face-centered cubic (fcc) type (Supplementary Fig. 2). The three-dimensional morphology of the film was examined by TEM, which reveals (Fig. 1a) that the film exhibits a typical columnar growth (Supplementary Fig. 3) with an average diameter of columnar grains (defined as grain size $d$) of ~11.1 nm, and there is a

segregation phase with a thickness of ~2.4 nm between the columnar grains (Supplementary Fig. 4), which can be described as a "thick grain boundary (TGB)"[30]. Figure 1b shows a high-resolution TEM (HRTEM) image of a grain and the surrounding TGB in plan-view. The standard Cu diffraction spots and amorphous halo are characterized by the Fast Fourier transform (FFT) in the grain and TGB, respectively (Fig. 1c, d). The measured crystal plane spacing (Fig. 1b) and positions of X-ray diffraction (XRD) peaks (Supplementary Fig. 2) indicate that the grain comprises Cu crystal in the fcc structure with almost no lattice expansion, indicating little boron solid solution in the Cu lattice. HRTEM image of the film in cross-sectional view (Fig. 1e) reveals that the columnar crystalline copper and amorphous boron regions appear alternately. An inverse FFT (IFFT) analysis is performed on different positions of the film to judge the dislocation distribution. The results (Fig. 1f, g and Supplementary Fig. 5) show that the distribution of dislocations in copper grains is mainly around the TGB. High-angle annular dark-field scanning transmission electron microscopy (HAADF-STEM) and energy-dispersive X-ray spectroscopy (EDS) analysis of the film in plan-view (Fig. 1h, i) reveals that Cu is concentrated in the grain. EDS (Fig. 1j, k) and electron energy loss spectroscopy (EELS) analysis (Supplementary Fig. 6) were performed on a double spherical aberration-corrected scanning transmission electron microscope (AC-STEM) in cross-sectional view. The analysis confirms the enrichment of B elements at the TGBs and the enrichment of Cu elements within the grains. XPS measurements (Supplementary Fig. 1) reveal there is no split peak, indicating a lack of Cu-B bonding and suggesting that Cu and B exist in the grain and TGB regions, respectively, in a segregated and immiscible state. These observations indicate that we have succeeded in constructing a "bamboo-like" columnar structure on the nanoscale, and such nanocolumnar structural units are connected to form a honeycomb network.

Within our "bamboo-like" dual-phase Cu-B nanocomposite film, there is a significant absence of interruptions in the columnar growth (Fig. 1 and Supplementary Fig. 5). It indicates that boron mainly influences the initial nucleation of Cu grains and increases the nucleation density, contributing to the grain refinement. Boron atoms fill the intergranular positions between the Cu grains to form a secondary phase that constitutes the TGB. During the columnar growth process, the two phases do not interfere with each other, resulting in the formation of elongated columnar grains. This phenomenon is driven by the immiscibility and non-bonding nature of copper and boron. Previous simulation and experimental work investigated the effects of diffusion kinetics and deposition parameters on the phase separation and nanostructure of binary thin films.[31–35] We calculated the formation energy of Cu-B substitutional and interstitial state, and the results are all positive (Supplementary Fig. 7a), indicating that Cu-B solid-solution structures are unstable. Moreover, we performed ab initio molecular dynamics (AIMD) simulations that initially place boron atoms in the substitutional and interstitial positions of the Cu lattice at 400 K (to simulate the increased temperature caused by the ion bombardment during the sputtering deposition process), and the results show that all boron atoms move to the surface of the Cu lattice (Supplementary Fig. 7b, c). Similar effects of GB segregation on grain refinement and structure stability were found in previous studies[36,37]. These assessments elucidate the observed phase segregation that produces grain refinement in the "bamboo-like" dual-phase nanocolumnar Cu-B nanocomposite.

To draw a comparative analysis, we also synthesized pure Cu, pure B and Cu-B films with varying B concentrations (Supplementary Table 1), namely lower (10.3 at.%) and higher (36.1 at.%), and subsequently subjected them to structural analysis. The B concentration is derived from the XPS measurements (Supplementary Fig. 1), which evince no discernible split peak. The aforesaid observation is congruous with the "bamboo-like" dual-phase Cu-B nanocomposite film under consideration, thereby signifying that the segregation tendency

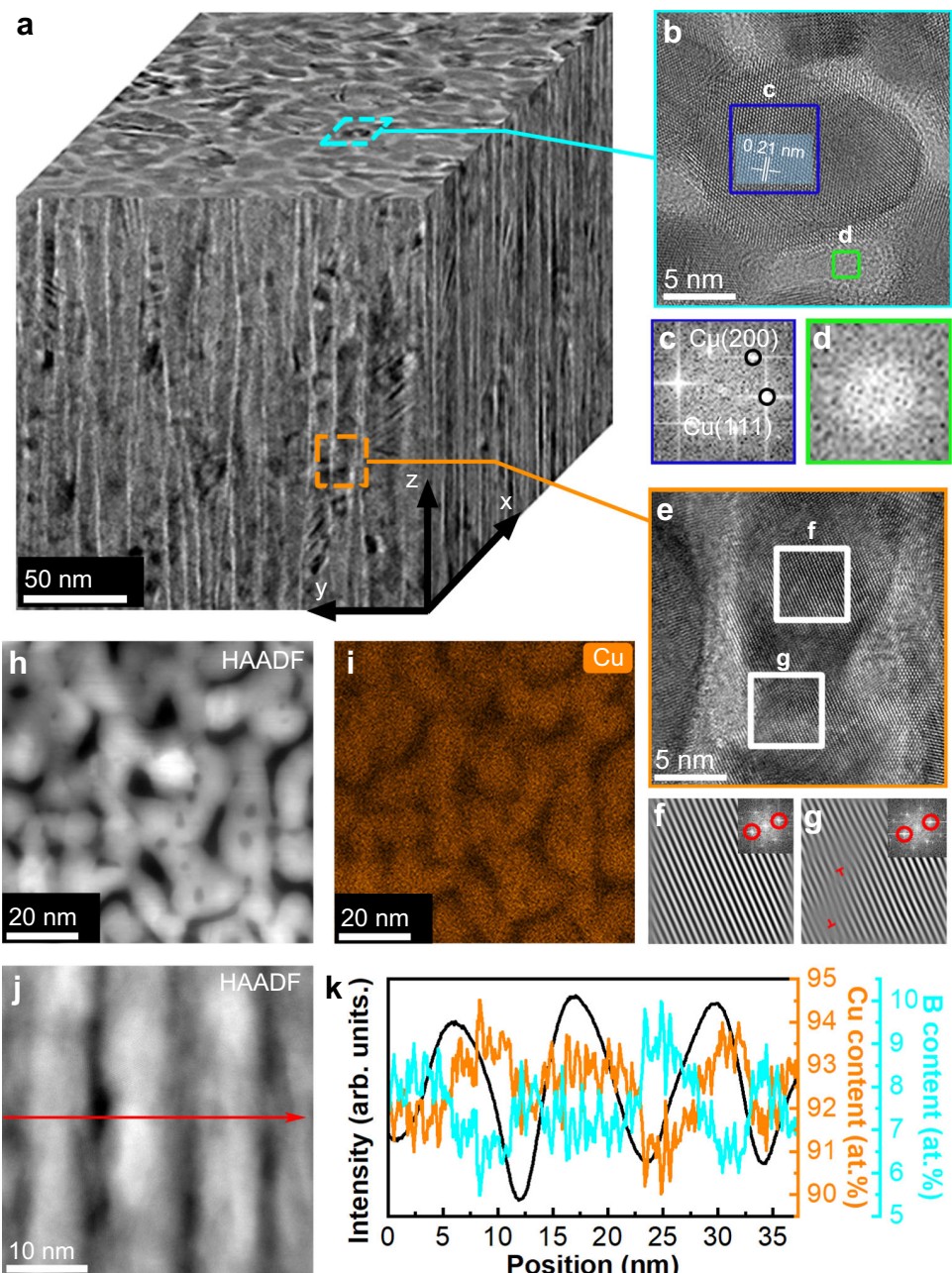

**Fig. 1 | TEM images of the structures of the "bamboo-like" dual-phase Cu-B nanocomposite film. a** The three-dimensional, reconstructed TEM image. The surface of the sample lies in the x−y plane, and the z axis indicates the depth from the surface. **b** Representative plan-view HRTEM image, which shows the structure of the amorphous region wrapping the crystalline region, and the distance between the crystal planes in the crystalline region is 0.21 nm. **c, d** The Fast Fourier transform (FFT) image of the crystalline region and the amorphous region, respectively. Diffraction spots on the Cu (111) and Cu (200) planes can be distinguished in (**c**). **e** Representative cross-sectional view HRTEM. Two regions were selected,

respectively in the grain interior and near the GB of the columnar crystal structure. **f, g** The inverse Fast Fourier transform (IFFT) image of the two selected regions in (**e**), respectively. The upper right panel shows the corresponding FFT and filtered (111) diffraction spots. The "⊥" symbols represent dislocations. **h** Plan-view HAADF-STEM image. **i** The element mapping of Cu corresponding to (**h**). **j** Cross-sectional view of the HAADF-STEM image. the red arrow shows the position and direction of the EDS line scan. **k** The contrast of the HAADF image (black), Cu content (orange) and B content (blue) as a function of position.

of Cu and B in the Cu-B system remains unaffected by the B concentration. The XRD results indicate that both the pure Cu and Cu-B films exhibit a fcc crystal structure, while the pure B film is amorphous (Supplementary Fig. 2). And the intensity of the peaks diminishes as the B concentration rises. The addition of element B causes a significant decrease in the grain size, from 18.9 nm for the pure Cu film to 8.1 nm for the Cu-10.3 at.% B film, 9.5 nm for the Cu-26.5 at.% B film, and 6.5 nm for the Cu-36.1 at.% B film. The grain size of Cu-B films does not show a regular pattern with increasing B concentration and is close to

the critical grain size for Hall-Petch effect failure (Supplementary Table 1). We have observed the cross-sectional morphology of the Cu-B films with varying B concentrations. The Cu-10.3 at.% B film shows a morphology without significant features and there is a diffuse distribution of amorphous segregated phases (Supplementary Fig. 8). Conversely, in the Cu-36.1 at.% B film, the fundamental columnar growth morphology endures, yet a discernible discontinuity exists in the columnar structure, and the progression of columnar grain growth becomes impeded by the amorphous B phase (Supplementary Fig. 9).

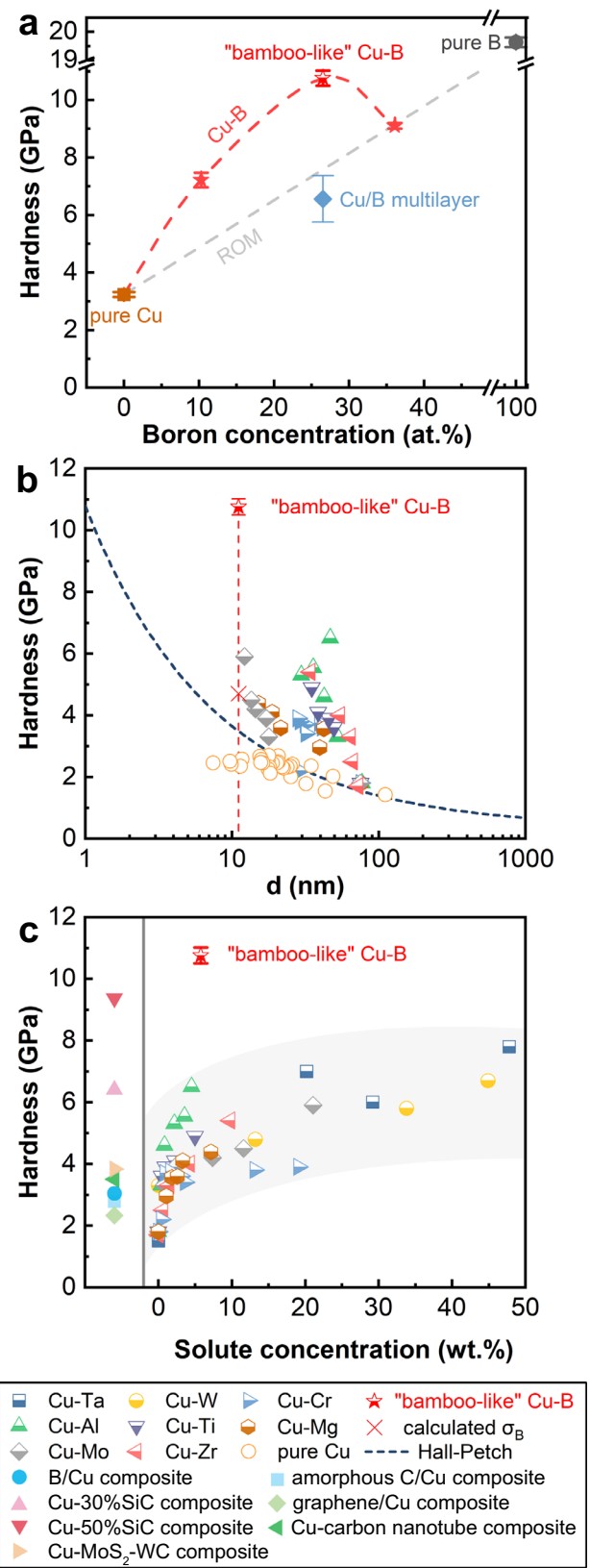

**Fig. 2 | Nanoindentation results on the "bamboo-like" dual-phase Cu-B nano-composite film. a** Hardness variation with boron concentration for all films synthesized. The gray dashed line is the predicted value based on the rule of mixing (ROM) for pure Cu and pure B. The error bars represent standard deviations. **b** Hardness variation with grain size (*d*) in logarithmic scale for the "bamboo-like" dual-phase Cu-B nanocomposite film compared with other binary Cu alloys. The black dashed line indicates the hardness of nanostructured Cu predicted by the Hall-Petch effect. The red cross marks the hardness obtained via a comprehensive evaluation of the boundary strengthening effect (see Supplementary Information). **c** Hardness variation with the solute element weight concentration for the "bamboo-like" dual-phase Cu-B nanocomposite film compared with other binary Cu alloys, and the hardness values of some Cu matrix composites are listed on the left for comparison. The same legends in (**b**) and (**c**) are shown in the bottom panel.

resembles that of the nano multilayer structure. To investigate the structure-property correlation, we synthesized a Cu/B multilayer film possessing similar structural dimensions. The multilayer is comprised of a crystalline Cu layer with a thickness of ~10.6 nm and an amorphous B layer with a thickness of ~3.6 nm. The Cu layer exhibited a polycrystalline structure with discernible GBs (Supplementary Fig. 10).

## Hardness and the strengthening mechanism

We performed nanoindentation tests on the synthesized films mentioned above in the continuous stiffness measurement (CSM) mode in order to determine their hardness (Fig. 2a). The results indicate that the "bamboo-like" dual-phase Cu-B nanocomposite film exhibits the largest hardness of 10.8 ± 0.3 GPa compared with other Cu-B films and exceeds the hardness of the Cu/B multilayer film. That is, the hardness of Cu-B films shows an increasing then decreasing trend with the increase of B concentration, while the trend of their moduli is consistent with the hardness (Supplementary Table 1). Furthermore, the hardness of the Cu-B system is predicted using the rule of mixing (ROM), taking into account the hardness values of pure Cu and pure B films, while the "bamboo-like" dual-phase Cu-B nanocomposite film demonstrates considerably higher hardness than this prediction. We analyzed several factors that may affect the hardness of the films, including residual stresses and grain size. Our measured results show that the films have small residual stresses which do not show a regular trend, indicating that the residual stresses have a negligible effect on the hardness. At the same time, Cu-B films with varying B concentrations have similar grain sizes, thus are not expected to have a notably different contributions to hardness (Supplementary Table 1).

The nanoindentation hardness value far exceeds those achieved in other reported binary Cu alloy films, including Cu-Cr[23], Cu-Ti[38], Cu-Mo[38], Cu-Ta[39], Cu-W[40], Cu-Al[41], Cu-Mg[41], and Cu-Zr[42], measured in terms of grain size (Fig. 2b) or alloying element content (Fig. 2c), and is even higher than some Cu matrix composites, such as B-Cu composite[43], Cu-30% SiC composite[44], Cu-50% SiC composite[44], Cu-MoS$_2$-WC composite[45], amorphous C/Cu composite[46], graphene/Cu composite[47] and Cu-carbon nanotube composite[48]. Moreover, this hardness value (10.8 GPa) is more than double the highest hardness (4.698 GPa) for microstructurally strengthened copper expected by the classic boundary strengthening analysis (see Supplementary Information). Previous study showed[49] that the hardness of pure copper with grain sizes in the range of 10 nm deviated downward from the trend predicted by the Hall-Petch effect (Fig. 2b), which is in stark contrast with the greatly enhanced hardness of the "bamboo-like" dual-phase Cu-B nanocomposite film.

To unveil the mechanism underlying the unusually high hardness of the "bamboo-like" dual-phase Cu-B nanocomposite film, we examine its structural changes under indentation and compare it with the Cu/B multilayer film of similar cross-sectional structure. Figure 3 shows the microstructural changes of the "bamboo-like" dual-phase Cu-B film after nanoindentation. A typical pile-up behavior appears in the scanning electron microscopy (SEM) image (Fig. 3a), which is also seen in

The above results indicate that only a suitable B concentration is conducive to the formation of the "bamboo-like" dual-phase nanocomposite structure, forming continuous and uniform amorphous B phases and maintaining the growth of columnar Cu grains without interruption. Moreover, it is observed that the cross-sectional morphology of the "bamboo-like" nanocolumnar structure closely

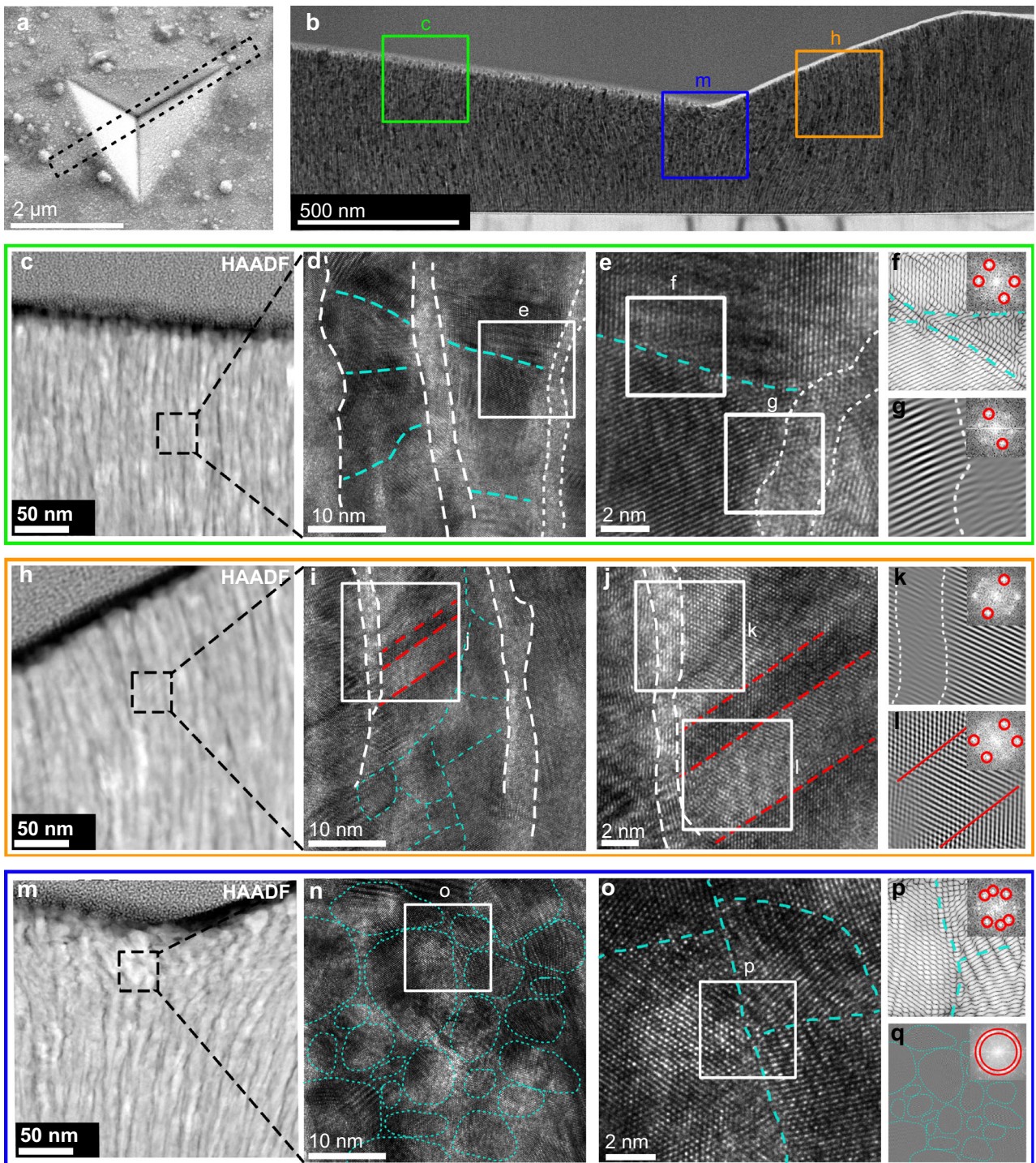

**Fig. 3 | Morphology of the indented "bamboo-like" dual-phase Cu-B nano-composite film. a** Representative indentation SEM image. The dashed line marks the location where FIB prepared the sample for the cross-sectional view. **b** The cross-sectional TEM images of Cu-B film after nanoindentation test, showing the deformation behavior. Three boxes mark the regions with different degrees of deformations. **c**–**g** HAADF-STEM, HRTEM, and IFFT images of the region with lower plastic strains. **c** The HAADF-STEM image shows columnar crystals with little bending. **d** The representative enlarged HRTEM image of the dashed box area in (**c**). White dashed lines represent the original GBs and blue dashed lines represent newly formed GBs after indentation induced deformation. **e** A magnified view of the white box in (**d**), which contains both the TGBs and newly formed GBs. **f**, **g** The IFFT images corresponding to the boxes in (**e**), which contain the newly formed GBs and TGBs, respectively. The upper right corners are the corresponding FFT images, and the red circles are the selected filtered (111) crystal plane diffraction spots.

**h**–**l** HAADF-STEM, HRTEM, and IFFT images of the region with higher plastic strains. **h** The HAADF-STEM image shows columnar crystals with large bending. **i** The representative enlarged HRTEM image of the dashed box area in (**h**). The red dashed lines represent the deformation induced twin boundaries. **j** A magnified view of the white box in (**i**). **k**, **l** The IFFT images corresponding to the boxes in (**j**), which contain the TGBs and deformation induced twin boundaries, respectively. **m**–**q** HAADF-STEM, HRTEM, and IFFT images of the region with the largest observed plastic strains. **m** The HAADF-STEM image shows refined grains after severe plastic deformation. **n** The representative enlarged HRTEM image of the dashed box area in (**m**). Blue dashed lines roughly indicate the change in the orientation of the lattice fringes. **o** A magnified view of the white box in (**m**). **p** The IFFT image corresponding to the box in (**o**), which contains trigeminal GBs. **q** The IFFT image of (**n**), which is used to identify the GBs in (**n**).

the 3D atomic force microscopy (AFM) image (Supplementary Fig. 11) of the residue indentation morphologies. No cracks appear near the indentation site. TEM morphology of the indentation cross-sectional view is shown in Fig. 3b. Three representative regions under different loading conditions are selected for an in-depth analysis, marked by the lower shear stress (Fig. 3c and Supplementary Fig. 12b), higher shear stress (Fig. 3h and Supplementary Fig. 12c), and complex stress (Fig. 3m and Supplementary Fig. 12d) loading conditions. HAADF-STEM images reveal that the distribution of boron is mostly concentrated in the TGBs, which is consistent with the results of the undeformed film (Supplementary Fig. 1), except in the region just below the indenter tip (Fig. 3m) where dispersed boron distribution indicates a notable indentation induced grain refinement. HRTEM analysis was performed to explore the deformation induced structural changes in the three selected regions. In the region with lower shear stress and deformation (Fig. 3d, e), the original columnar grains and TGBs undergo small bending along with the formation of GBs, like a diaphragm at the bamboo node, whereas the similar region of the Cu/B multilayer film does not produce any bending phenomenon and only a slight decrease in layer thickness (Supplementary Fig. 13b, c). Moreover, the IFFT image (Fig. 3f) shows that the transition of lattice fringes at GBs presents a relatively disordered region, which is the result of dislocation movement. In the region containing both grain and TGB, the blurred and chaotic TGB region can be seen in the image obtained by IFFT based on the lattice fringe in the direction of (111) inside the grain (Fig. 3g). In the region with higher shear stress and deformation (Fig. 3i, j), the bending of columnar grains becomes more notable, and the density of GBs increases further, while some twin boundaries appear (Fig. 3l), indicating simultaneous occurrence of significant partial dislocation movement and grain rotation. In contrast, the multilayer film sprouts a shear band across the film thickness in the region of larger deformation (Supplementary Fig. 13f, g), with obvious faults on both sides of the shear band (Supplementary Fig. 13h, i) but without the phenomenon of layer bending. It should be noted that nano multilayers with layer bending during deformation are usually accompanied by shear bands[50–54], and the deformation mechanism is significantly different from the column structure bending of the "bamboo-like" dual-phase Cu-B nanocomposite film. In the region just below the indenter tip, the complex stress conditions produce significant grain refinement (Fig. 3n, o), producing a distinct GB morphology. IFFT images clearly distinguish random orientation changes of the (111) lattice fringes (Fig. 3p, q). Similar changes are produced in Cu/B multilayers. Overall, the IFFT images (Fig. 3f, g, k, l, p, q) reveal that defects in the grains are mainly concentrated around GBs, and a large number of disordered structures exist in the TGBs, which serve as the dislocation source and sink.

From the above analysis of structural changes under indentation and structural comparisons with the Cu-10.3 at.% B, Cu-36.1 at.% B and the Cu/B multilayer films, we conclude that the unusually large strengthening and the resulting improved hardness of the "bamboo-like" dual-phase Cu-B nanocomposite film mainly stems from the following three factors: (1) an indentation induced grain refinement, (2) a strong support of the TGBs consisting of an amorphous boron framework, (3) an enhanced stress response of the nanocolumnar copper structure constrained by the TGBs. Below, we present an in-depth assessment of each of these factors.

## Strengthening by indentation induced grain refinement

Grain size is a characteristic feature of nanocrystalline materials that often serves as a key parameter for evaluating GB induced strengthening (see Supplementary Information). In the present case of Cu-B nanocomposite film, the grain size is measured by the average columnar grain diameter, while the elongated columnar length is much larger. Under small deformation (Fig. 3c), we observed indentation induced formation of GBs inside the columnar grains (Fig. 3d, e),

which is driven by dislocation and GB motion that underpins the Hall-Petch effect.

Grain refinement caused by plastic strain in metals is a well-known phenomenon, especially in processes at large strains and high strain rates, such as ball milling and surface mechanical grinding treatment, which can refine grains to nanoscale. The mechanism behind grain refinement under large strains is widely attributed to the dislocation model, which posits that a dislocation cell structure forms during the initial stages of plastic deformation and gradually evolves into the final fine grain structure. In the refinement process, there is an accumulation of misorientation between neighboring dislocation cells[55]. Therefore, grain refinement at small strains is relatively rare because dislocation does not have sufficient motion to form GBs. However, in our nanocolumnar structure, the TGBs act as an excellent dislocation source, whereas the elongated nanocolumnar grains contains initial dislocations near the TGBs (Fig. 1 and Supplementary Fig. 5). Under stress excitations, dislocations nucleate at and propagate along the TGBs. After a short period of movement, neighboring dislocations aggregate in grains and form new GBs with the bending of the columnar grains (Fig. 3).

In the HAADF-STEM image of the region directly below the indenter tip that undergoes the most severe deformations (Fig. 3m), it is seen that a small amount of boron appears between the refined grains, indicating that amorphous boron continues to move at the boundaries during grain refinement induced by indentation. Amorphous structures host high fluidity under the shear-band interactions and tend to deform in coordination with crystal grains[14,56,57]. GB movement and grain rotation during indentation deformation are thus inevitable, while amorphous boron flows into boundaries, hindering further movement of the grains thus strengthening the structure.

## Strengthening by amorphous boron framework TGBs

A prominent feature of the dual-phase Cu-B nanocomposite film is its TGBs comprising an amorphous boron framework, which contributes to enhancing the film strength and hardness beyond standard GB strengthening. In heterogeneous metallic systems, deformation process is closely related to GB complexion[10,30,58], which limits dislocation propagation by the ledges and solute atoms at the grain-complexion interface, leading to strong dislocation pinning and increasing the flow stress required for dislocation movement. TGBs also can effectively prevent dislocation propagation by acting as a dislocation sink[30].

To assess the role of the amorphous boron TBGs during indentation deformation, we tested the rate-limiting process of the Cu-B film compared with pure copper to extract the strain rate sensitivity (SRS) index $m$. The results (Fig. 4) show that the load-displacement curves shift to the left with increasing strain rate in both pure Cu and the Cu-B films, but to a lesser extent in the latter (Fig. 4a, b). The SRS index $m$ calculated from the slope of hardness-strain rate log-log plots (Fig. 4c) decreases from $0.050 \pm 0.019$ for pure Cu to $0.012 \pm 0.002$ for the Cu-B film. This variation of $m$ is attributed to the amorphous boron framework TGBs after excluding the influence by the grain size effect (see Supplementary Information). The deformation mechanisms for crystalline metals and amorphous solids are different in that the former is dominated by dislocation and GB movement, while the latter is mainly decided by the formation of shear bands and microcrack propagation. In general, the value of SRS decreases significantly, may even turn negative, when an amorphous phase is present[59]. In the dual-phase Cu-B nanocomposite film, when shear bands or microcracks form in the intergranular amorphous boron layer, the associated volume expansion is bound to run into the Cu grains, and due to the co-deformation between the distinct phases of copper grains and amorphous boron, the blockage by the Cu grains would turn the deformation of the amorphous boron into a flow-like process, with no obvious yield point or pop-in point as seen in the load-displacement curve (Fig. 4b). Overall, the amorphous boron that forms the TGBs not only impedes

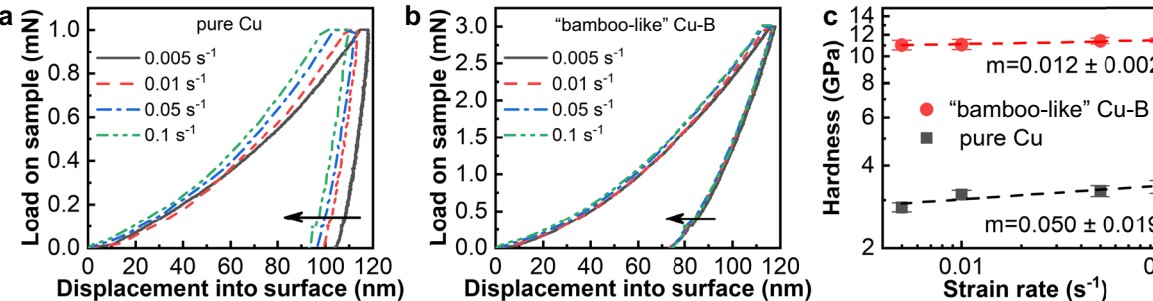

**Fig. 4 | Strain rate sensitivity of pure Cu and the "bamboo-like" dual-phase Cu-B nanocomposite film. a**, **b** Representative load-depth curves of pure Cu and the "bamboo-like" dual-phase Cu-B nanocomposite film obtained by load control mode, respectively. The arrow in each panel indicates the direction of the gradual increase in strain rate. **c** *log* (hardness)-*log* (strain rate) plots of pure Cu and the "bamboo-like" dual-phase Cu-B nanocomposite film. The slope of each line represents the strain rate sensitivity (*m*). The error bars represent standard deviations.

dislocation movement within Cu grains, but also contributes to the strengthening of the film due to its inherent high strength.

## Strengthening by enhanced stress response of constrained nanocolumnar copper

During nanoindentation tests, the material under the indenter experiences a biaxial stress that comprises a lateral shear and a normal compression component, and the ultimate strength under the compression constrained shear strains usually dictates the initiation of dislocations, leading to incipient plastic deformation and limiting the measured hardness. This process, however, may be drastically altered in materials possessing high structural anisotropy, such as the Cu-B nanocomposite film. Here, the nanocolumnar copper structure reinforced by the amorphous boron TGBs is highly resistant to lateral shear strains due to the hindrance by the TGBs, and as a result undergoes a predominantly bending deformation as revealed by the HAADF-STEM, HRTEM, and IFFT images of the indented regions shown in Fig. 3. A related scenario was recently reported by Guo et al.[60] who found that a GB constraint transforms the lateral shear stress in the (001) plane into the vertical sliding compression of columnar grains in $TiB_2$ films with a (001) texture, and the hardness was dictated by the [001] compressive strength, instead of the usual lateral shear strength in the (001) plane. This microstructurally constrained deformation mechanism explains the notably enhanced indentation hardness of the $TiB_2$ film. In the present work, we show direct experimental evidence of a bending deformation of the "bamboo-like" nanocolumnar Cu-B film, which brings about a distinct mechanism for resistance to indentation loading compared to the standard shear dominated structural response as schematically illustrated in Fig. 5a. To assess and compare pertinent mechanical responses of the columnar copper structure under different deformation modes, we calculated shear stress–strain relations in the (111) plane and compressive stress–strain relation in the [111] direction of fcc Cu crystal to assess the stress responses underlying the processes that determine the measured hardness (Fig. 5b, c). After checking the dynamic stability by phonon dispersion calculations (Supplementary Fig. 14), we obtained maximal shear stresses in the three high-symmetry crystallographic directions of (111)[-1-12], (111)[-101] and (111)[11-2] of 8.63 GPa, 3.02 GPa, and 2.20 GPa, respectively. Meanwhile, the compressive stress calculations produced an ultimate stress of 25.27 GPa. These results clearly show that copper has much weaker shear strength compared to its compression strength, and the latter is the main deformation mechanism of the columnar copper structure under bending, which contributes to the large strengthening compared to pure copper and its binary alloys that undergo the usual shear dominated deformation under indentation in the absence of the nanocolumnar structural constraint. It is worth noting that when the loading direction is perpendicular to the thin film, there exists a strong correlation between the structure and the loading. In analogy to the

multilayer configuration featuring a planar two-dimensional arrangement perpendicular to the loading direction, the bamboo-like structure exhibits a three-dimensional columnar morphology aligned in parallel with the loading direction. Enhancing the ability to suppress shear can effectively improve strength. Although multilayer structures can enhance strength by directly impeding dislocation motion at layer interfaces, bamboo-like structures can achieve better resistance to shear processes through improved bending response, thus demonstrating improved performance.

Based on the previous analysis, it is speculated that the potential to restrict the shear behavior of the "bamboo-like" dual-phase nanocomposite structure may impede material failure arising from shear deformation, thereby making a noteworthy contribution toward enhancing ductility. To verify this scenario, we conducted in situ compression tests on the film, as displayed in the Supplementary Movie 1. The results indicate that the engineering stress–strain curve during the testing process is remarkably smooth and does not exhibit any pop-in points (Fig. 6a). On the premise that the deformation of the micropillar is uniform, the true stress–strain curve is derived and show a yield strength ($\sigma_{0.2\%}$) of ~1.64 GPa and a flow stress ($\sigma_{max}$) of ~2.45 GPa (Fig. 6b). However, the deformation process of the micropillar is non-uniform, primarily dominated by barrel-shaped deformation at the top. Hence, by refitting the true stress–strain curve using the real-time measurement of the micropillar's cross-sectional area (Fig. 6c), we obtained a yield strength of ~1.36 GPa and a flow stress of ~2.58 GPa. Notably, the micropillar exhibited no shear bands or cracks even at strains exceeding 50%, indicating high plasticity and ductility. In contrast, Cu-based micropillars reported in previous studies with similar strength did not exhibit such high ductility[61–64]. The results confirm our hypothesis that the "bamboo-like" dual-phase nanocomposite structure constrains the shear behavior, resulting in high strength and ductility. It signifies that the "bamboo-like" dual-phase Cu-B nanocomposite film is substantially hardened while retaining the intrinsic ductility of the metal, thereby achieving remarkable strengthening and toughening.

In this work, we designed and constructed a "bamboo-like" nanocolumnar copper structure reinforced by an amorphous boron framework that serves as a strong and robust thick grain boundary network. The combination of copper and boron is chosen based on the nearly complete immiscibility of these two elements, which is favorable for the formation of the desired dual-phase segregated nanocolumnar film structure that was synthesized via the magnetron sputtering co-deposition technique. The distinct structural features of the Cu-B nanocomposite film dictate a bending mode, instead of the usual compression constrained shear mode, as the main deformation mechanism in response to the indentation loading. Combined with the strengthening by grain refinement and the strong amorphous boron framework, this unique mechanism leads to greatly enhanced

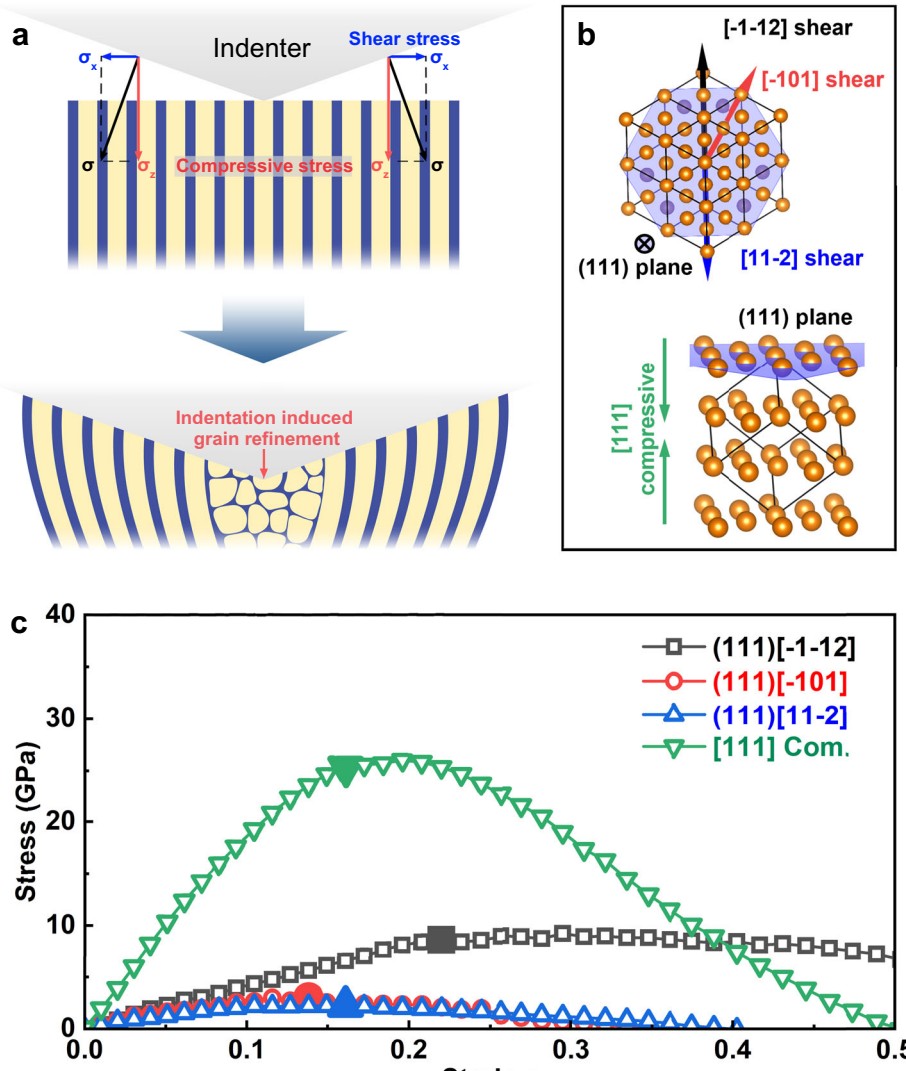

**Fig. 5 | Modeling and calculation of mechanical responses of the "bamboo-like" dual-phase Cu-B nanocomposite film. a** Schematic illustration of the mechanical response of dual-phase Cu-B nanocolumnar structure under indentation, which produces the stress conditions that can be decomposed into coexisting compressive and shear stresses, as shown by the red and blue arrows, respectively. The nanocolumnar Cu grains and amorphous boron TGBs are represented by the yellow and blue regions, respectively. As the indenter presses into the sample, the microstructure of the sample changes as indicated. In the region directly below the indenter tip, the film undergoes indentation induced grain refinement. The overall structural changes under indentation are dominated by the bending deformation of the nanocolumnar grains. The confinement by the boron TGBs severely restricts the shear deformation normally induced by indentation, leading to the predominantly bending mode of the nanocolumnar copper as the main structural change, generating enhanced stress responses. **b** Crystal structure of Cu at equilibrium with the directions of the shear and compressive strains indicated by the arrows. **c** Calculated stress responses of Cu under the shear strains in the (111) plane along the major high-symmetry [11-2], [-1-12] or [-101] shear slip directions and the compressive strain in the [111] direction. Enlarged symbols mark the highest stress values right before the onset of dynamic instability determined by the phonon dispersion calculations (Supplementary Fig. 14).

nanoindentation hardness of 10.8 GPa, while maintaining excellent strength (yield strength of ~1.36 GPa and flow stress of ~2.58 GPa) and ductility (failure strain of over 50%). These results offer fresh insights for conceptual design and establish effective synthesis routes for experimental implementation of greatly strengthened metals via anisotropic heterostructure construction using immiscible metal-light element combinations. This strategy may work for structural construction and performance enhancement in a variety of metal films for advanced instrument and device applications.

## Methods
### Materials
The "bamboo-like" dual-phase Cu-B nanocomposite film was deposited on Si(100) substrates by magnetron co-sputtering at room temperature. Pure Cu (99.999%) and B (99.95%) targets were used to prepare the ~550 nm-thick films. The sputtering chamber was evacuated to a base pressure below $5 \times 10^{-4}$ Pa, and a 0.8 Pa Ar (99.999%) pressure was maintained during the deposition process via controlling the flow rate of Ar at 80 sccm. The substrates were ultrasonically cleaned in acetone and ethanol and blown dry before deposition. The substrate was neither heated nor cooled with the rotational speed of 5 r/min, and the distance between the target and the substrate was 8 cm. During the deposition, the substrate bias was −80 V, the direct current (DC) power of the Cu target is 20 W, and the radio frequency (RF) power of the B target is 250 W. Pure Cu and pure B films with similar thicknesses were synthesized using only Cu target and B target sputtering, respectively, while the rest of the deposition parameters remained constant. Cu-B films with varying B concentrations were synthesized by changing the B target power to 150 W and 350 W, respectively, while the rest of the deposition

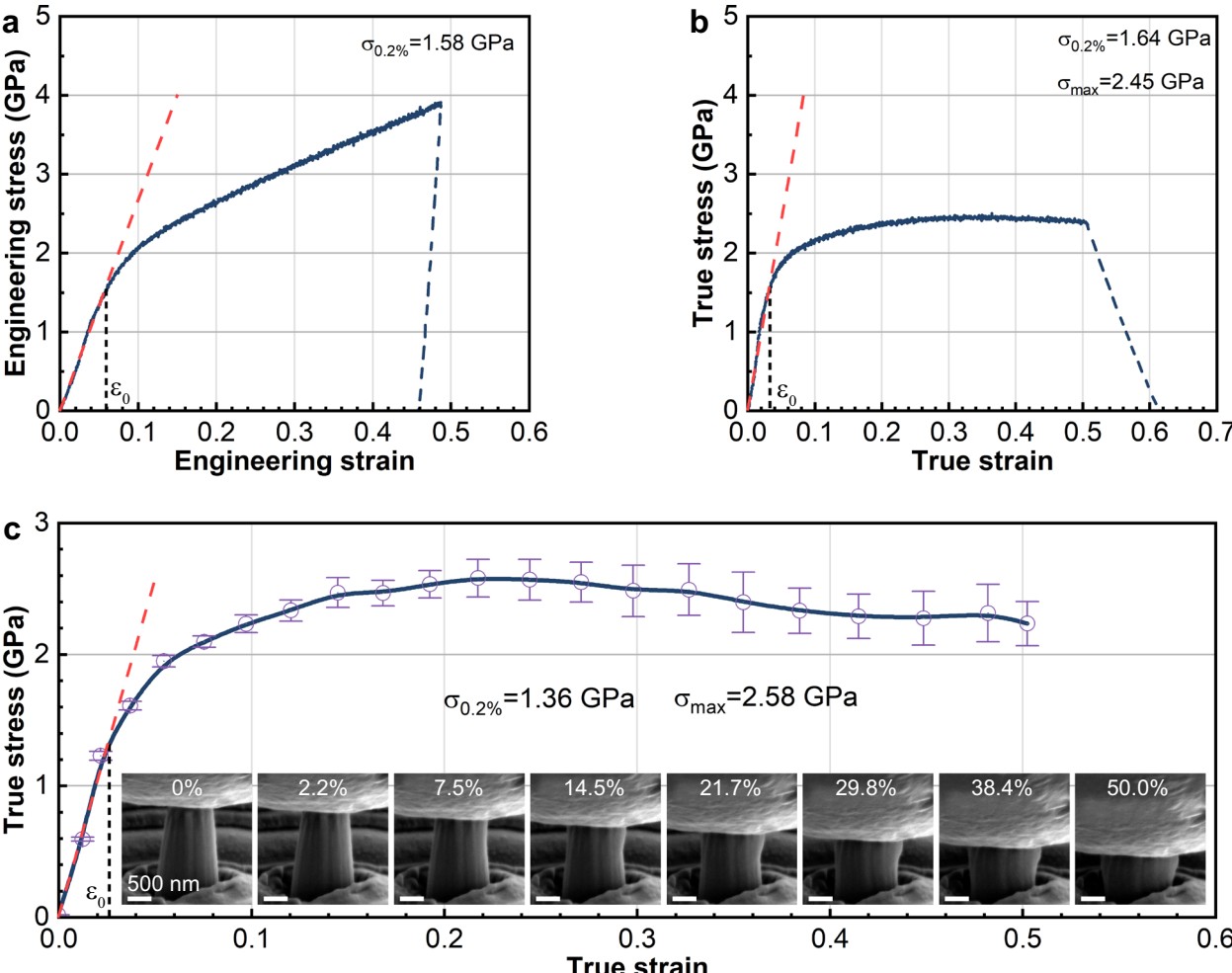

**Fig. 6 | Results of in situ compression tests of the "bamboo-like" dual-phase Cu-B nanocomposite film. a** Engineering stress–strain curve. The red dashed line is a linear fit to the elastic regime, where the micropillar shows elastic behavior until the first yield point ($\sigma_0$) at a strain ~0.06 ($\varepsilon_0$). The yield strength $\sigma_{0.2\%} = 1.58$ GPa corresponds to the stress at a strain of $\varepsilon_0 + 0.2\%$. **b** True stress–strain curve obtained assuming uniform micropillar deformation. The yield strength $\sigma_{0.2\%} = 1.64$ GPa, while the flow stress $\sigma_{max} = 2.45$ GPa corresponds to the stress at a strain of $\varepsilon_0 + 8\%$. **c** True stress–strain curve obtained by fitting the real-time measurement of the micropillar cross-sectional area. The yield strength $\sigma_{0.2\%} = 1.36$ GPa, while the flow stress $\sigma_{max} = 2.58$ GPa. Furthermore, the SEM images of the micropillar at different strains are also provided in the figure. The error bars represent standard deviations.

parameters remained constant. For the Cu/B multilayer film, the Cu target power was 20 W and the B target power was 250 W. Both targets were sputtered alternately with a sputtering time of 2.5 min and 30 repetitions. The consistent target power with the "bamboo-like" dual-phase Cu-B nanocomposite film and equal sputtering time per layer ensure that the elemental composition of the Cu/B multilayer film is consistent with that of the "bamboo-like" dual-phase Cu-B nanocomposite film. We determined a sputtering time of 2.5 min/ layer so that the Cu/B multilayer film consists of ~10.6 nm-thick Cu layer and ~3.6 nm-thick B layer alternating with the top and bottom layers of Cu, which is the comparable structural dimension of the "bamboo-like" dual-phase Cu-B nanocomposite film. The sample used for the in situ compression tests was a 2.4 μm-thick Cu-26.5 at.% B film (B-target power of 250 W) prepared by extending the sputtering time to 8 h only.

### Structural characterization

X-ray diffractometer (XRD, D8-tools-Bragg-Brentano) with Cu Kα radiation at room temperature was used to determine the crystallographic orientations of the films. The bonding state of the films was analyzed by X-ray photoelectron spectroscopy (XPS, PerkinElmer PHI-5702) using monochromatic Al Kα radiation as the X-ray source at

1 keV energy. Before the test, the film surface was etched with Ar⁺ at 2 KeV energy for 5 min and 500 eV for 15 min to remove surface contaminants and reduce the probability of B being sputtered out. Microstructures and element distribution of the films were examined by using high-resolution transmission electron microscopy (HRTEM), high-angle annular dark-field scanning transmission electron microscopy (HAADF-STEM), dark-field scanning transmission electron microscopy (DF-STEM), and energy-dispersive spectroscopy (EDS) on a FEI Talos F200X TEM with an accelerating voltage of 200 kV. EDS and electron energy loss spectra (EELS) under the double spherical aberration-corrected scanning transmission electron microscope (AC-STEM, Thermo Scientific™ Themis Z) were used to further characterize the elemental distribution. The TEM samples were prepared using focused ion beam (FIB) scanning electron microscope (FEI Strata 400S). Prior to FIB milling, a thin Pt layer was deposited to protect the specimen from ion irradiation damage. The selected area was cut down perpendicular to the sample surface and removed, and the removed part was milled repeatedly to achieve the thickness that can be observed by TEM. The plan-view sample was first prepared by a fine mechanical polishing of the cross section to achieve a flat surface before proceeding with the above procedure. The surface morphology of indentation was measured using scanning electron microscope

(SEM, JEOL JSM-6700F) and atomic force microscope (AFM, Dimension Icon, Veeco Instruments/Bruker, Germany).

## Mechanical tests and characterization

Mechanical properties of the films were measured using an Agilent Nano Indenter G200 with a standard Berkovich tip at room temperature. The continuous stiffness measurement (CSM) mode was used to evaluate the hardness and the sustained mechanical response under indentation. Throughout the indenter loading process, a minor dynamic oscillation with a frequency of 45 Hz was introduced to the displacement signal. This allowed the amplitude and phase of the corresponding force signal to be captured using a frequency-specific amplifier, enabling a continuous measurement of stiffness. By continuously measuring hardness, it becomes possible to derive a continuous function of hardness and depth from the indentation experiment. The hardness value is then calculated from the hardness-displacement curve. To ensure accurate results and avoid surface and substrate effects, the range of hardness values is typically chosen to be ~10% of the film thickness. The maximum depth was 400 nm, and the hardness for the films was taken at a depth of ~50–60 nm. The Quasi-Static nanoindentation test with a maximum depth of 55 nm was used to further verify the accuracy of hardness measurement. In the quasi-static nanoindentation test, the hardness $H$ is calculated from the loading-unloading curve by the formula[65] $H = P_{max}/A$, where $P_{max}$ is the maximum load obtained from loading-unloading curve, and $A$ is the contact area, which is calculated by the empirical formula $A = C_0 h_c^2 + C_1 h_c + C_2 h_c^{1/2} + C_3 h_c^{1/4} + C_4 h_c^{1/8} + C_5 h_c^{1/16}$, where $C_0$ is a constant equal to 24.7 for a Berkovich tip, $C_1 - C_5$ can be obtained from the experimental data, and $h_c$ is the contact depth given by $h_c = h_{max} - \varepsilon P_{max}/S$, where $h_{max}$ is the maximum depth, $\varepsilon$ is a constant related to the indentation tip shape ($\varepsilon = 0.75$ for Berkovich), and $S$ is the contact stiffness, which is the initial unloading slope at the unloading stage of the loading-unloading curve. The load-controlled mode was used to evaluate the rate-limiting process of the films. The strain rates ($\dot{\varepsilon}$) is calculated by the loading rate ($\dot{P}$)[66]: $\dot{\varepsilon} = \frac{\dot{P}}{2P}$, where $P$ is the load and $\dot{P} = dP/dt$. And we set up four different strain rates: 0.005, 0.01, 0.05, and 0.1 s$^{-1}$. The strain rate sensitivity is experimentally defined as the slope of the double logarithmic plot of hardness $H$ and $\dot{\varepsilon}$ under isothermal conditions, which can be expressed as $m = \frac{\partial \log(H)}{\partial \log(\dot{\varepsilon})}$. The holding time for all the tests was 10 s to reduce the effect of creep, and the unloading rate was consistent with the loading rate. To reduce the effect of random measurement error, we made at least nine indentations at different places on the film surface for each sample, and the distance of each indentation was set at 30 μm. For CSM sample, the number of indentations was more than 20. In situ compression experiments were conducted employing a Hysitron TI950 nanoindenter equipped with a 5-μm diamond flat punch, utilizing a displacement-controlled mode at an approximate rate of 2 nm/s. For the uniaxial micro compression tests, micropillars were fabricated using an FIB instrument (FEI Strata 400S). The height of the pillars was ~2 μm, and their aspect ratios (height/diameter) were 2.

## First-principles calculations

The total-energy and stress–strain relation calculations were performed using the Vienna Ab initio Simulation Package (VASP) code[67]. The exchange correlation was described by the Perdew-Burke-Ernzerhof (PBE) functional[68] under the generalized gradient approximation (GGA) using the projector-augmented wave formalism[69]. A cutoff energy of 400 eV and an $8 \times 8 \times 8$ Monkhorst–Pack k-point grid[70] were used for fcc Cu with the $2 \times 2 \times 2$ supercell containing 32 Cu atoms to ensure that all the enthalpy calculations were converged to better than 1 meV/atom. The stress–strain relationships were determined using a quasistatic loading method, in which the lattice vectors were gradually deformed in alignment with the applied strain. Throughout the structural deformation process, the applied strain

along the designated loading path was kept constant to derive the corresponding stress response. Simultaneously, the relaxation of the remaining five independent components of the strain tensors and all atoms within the unit cell was conducted until the residual components of the Hellmann-Feynman stress tensor orthogonal to the applied strain reached values lower than 0.1 GPa. Additionally, the force acting on each atom was minimized to a negligible level. This approach with a relaxed loading path has been successfully applied to the calculation of the strength of several metals[71,72]. The shape of the unit cell is determined by the full atomic relaxation without any imposed boundary conditions. We calculated the compressive stress in the Cu [111] direction and also determined the shear stress in the easy-slip (111) plane and obtained the critical shear stresses. The lattice dynamical properties were calculated via the density functional perturbation theory (DFPT) method as implemented in the Phonopy code[73] to examine the dynamic stability of the strained structures.

## Data availability

All data are available in the manuscript and the Supplementary Information. All other data are available from the corresponding author upon request.

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

## Acknowledgements

This work was supported by the High-Performance Computing Center of Jilin University, China. National Natural Science Foundation of China grant Nos. 52322206, 51972139 (K.Z.), China Postdoctoral Science Foundation grant No. 2020M681031 (C.L.), Science and Technology Development Program of Jilin province grant No. 20210101062JC (K.Z.).

## Author contributions

K.Z. conceived this project. H.L. and X.H. performed the experiments. X.G. and C.L. carried out the theoretical calculations and analysis. H.L., X.G., C.L., M.W., Z.W., C.C. and K.Z. discussed the results. C.L., K.Z., W.Z. and C.C. supervised and coordinated the work. H.L, K.Z., C.L. and C.C. wrote the manuscript with input from all authors.

## Competing interests

The authors declare no competing interests.
