## [Peer Review File · Nature Communications]

Bamboo-like dual-phase nanostructured copper composite strengthened by amorphous boron frameworkREVIEWER COMMENTS

Reviewer #1 (Remarks to the Author):

This manuscript presented an interesting study on the mechanical behavior of CuB coatings. It is remarkable that their sputtered CuB coatings have hardness exceeding 10 GPa. In comparison, most Cu and Cu alloy coatings have hardness well below this value. While the results are exciting, the followings should be addressed before the manuscript can be considered for publication.

1. EDS map for Cu-B coating has been provided. However, EDS line profiles should also be presented so that one can see the Cu composition modulation, especially across the amorphous phase boundaries.
2. It is known that residual stresses develop in sputtered films. The authors shall quantify residual stresses in these Cu-B alloy films.
3. The elastic modulus of the Cu-B films should also be provided.
4. Also have the authors looked at Cu-B alloy films with different B composition? A set of Cu-B films with variable composition will help to establish the solid trend showing the influence of B composition and grain size on mechanical behavior of the Cu-B coatings.

Reviewer #2 (Remarks to the Author):

The manuscript by Lv et al, "Nanostructured copper with amorphous boron decorated boundaries exhibits record-high hardness," suggests that the unique bamboo structure created by co-deposition using magnetron sputtering underlies the high hardness of this copper alloy. The work not only reports on the hardness, but also includes detailed STEM observation of the deformed material to relate the deformation mechanism to the structure. The work also includes ab initio molecular dynamics simulation to examine how the deposition process of Cu-B leads to the bamboo structure. Lastly, the authors perform DFT calculation for the solubility of B in Cu. In total, this is a very nice paper. I do not believe, however, that the authors have shown that underlying cause for the high hardness in their Cu-B material is fundamentally different from the nano crystalline-core/amorphous-shell structure explanation (i.e.-supra-nanometre-sized dual-phase glass-crystal) provided in their ref [14] for a Mg based alloy. Possibly the authors could have better related hardness to structure by varying the B concentration, or by showing that a Cu/B nanolaminate structure with comparable layer thicknesses and compositions has significantly lower hardness. Lastly, the authors only provide information about the hardness, but very little regarding other mechanical properties, thermal stability, or potential applications.

Reviewer #3 (Remarks to the Author):

The authors present a manuscript on the hardness / strength and deformation mechanisms of a copper -

boron nano composite. Beside nano indentation extensive use of electron microscopy is made to identify the microstructure and changes due to plastic deformation by indentation. They found a very high hardness 10.8 GPa and explained this by 3 main processes: (i) indentation induces grain refinement, (ii) strong support of the boron network and (iii) enhanced stress response of the columnar copper grains due to the constrain of the boron network.

In the manuscript the authors mention often their material as copper alloy. In my opinion this is misleading as the structure of the current material is a nano composite consisting of columnar copper grains with a diameter of about 11 nm and an amorphous boron network at the "grain boundaries" with a thickness of about 2.5 nm. Hence a comparison with copper alloys is not useful. Overall, the structure and also the deformation processes looks similar to that of nanoscale multilayers. In fact, I am missing here some literature comparison of hardness and deformation patterning (e.g. bending of the layers).

The description of the deformation processes is a little bit speculative. All the mentioned processes make sense and it is likely that they contribute to the hardness increase, however, their contribution is not quantified or discussed which one is maybe the most important.

A quick literature review (e.g. Atomic arrangement and mechanical properties of chemical-vapor-deposited amorphous boron, Jessica M. Maita, Gyuho Song Mariel, Colby Seok-Woo Lee, *Materials & Design*, Vol. 193, August 2020 or Amorphous boron coatings produced with vacuum arc deposition technology, C. C. Klepper, R. C. Hazelton, E. J. Yadlowsky, E. P. Carlson, and M. D. Keitz, *Journal of Vacuum Science & Technology A* 20, 725 (2002)) shows that the nano hardness of amorphous boron thin films is between 30 and 35 GPa. If one assumes an area fraction of the amorphous boron of about 25 to 30% (see Fig. 1A top view where the indentation is performed or Fig. 1H) the measured hardness with 10.8 GPa can be explained by a simple parallel composite model. Hence, all the other discussed mechanisms are not necessary. A more in-depth discussion is needed here to justify the the proposed mechanisms and what contribution to hardness they really have. Also tensile properties (ductility) and fracture toughness would be of interest (I know this was not the scope of the current work, however, for a "good" material these parameters are also essential).

Overall, the manuscript presents some interesting work on the production of a Cu-B nano composite which can maybe extended to other material combinations. On the other side, the explanations of the so-called "record-high" hardness are insufficient. In my opinion, the hardness is not really "record high" and can be explained by the high hardness of the amorphous boron network by a simple composite model. Also, I am missing the interesting comparison with nanostructured multilayer thin films because some of the mentioned deformation mechanisms are also applying there.

I think the manuscript doesn't meet the high standard of Nature Communication and I don't recommend publication.

Manuscript ID: **NCOMMS-22-48845**

Title: **Nanostructured copper with amorphous boron decorated boundaries exhibits record-high hardness**

Authors: Hang Lv, Xinxin Gao, Kan Zhang, Mao Wen, **Xingjia He**, Zhongzhen Wu, Chang Liu, Changfeng Chen, Weitao Zheng

Dear Reviewers:

Thank you for your communication concerning our manuscript referenced above. We are pleased to see that the reviewers have made mostly positive assessments about our reported work along with constructive comments and suggestions for improving the presentation and discussion of the manuscript. In response, we have performed additional experiments and analyses to address all the issues raised by the reviewers. Based on the obtained results and related considerations, we have prepared a detailed point-by-point response (attached below) to all the reviewer comments and suggestions. We also have made corresponding changes in the revised manuscript with the updated contents marked in red for the convenience of reading. In addition, we have put additional supporting material to the Supplementary Information.

Yours Sincerely,

Kan Zhang,

Professor,

Materials Science and Engineering,

Jilin University, Changchun, China

E-mail: kanzhang@jlu.edu.cn

Responses to the comments by reviewers

Reviewer #1: This manuscript presented an interesting study on the mechanical behavior of Cu-B coatings. It is remarkable that their sputtered Cu-B coatings have hardness exceeding 10 GPa. In comparison, most Cu and Cu alloy coatings have hardness well below this value. While the results are exciting, the followings should be addressed before the manuscript can be considered for publication.

Authors' reply: We appreciate the reviewer's positive appraisal of our work.

Comment 1:

1. EDS map for Cu-B coating has been provided. However, EDS line profiles should also be presented so that one can see the Cu composition modulation, especially across the amorphous phase boundaries.

Authors' reply: We thank the reviewer for raising this issue. Following the reviewer's recommendation, we performed EDS and EELS examinations utilizing AC-STEM. We have added the following content in the revised manuscript: "EDS (Figs. 1J and 1K) and electron energy loss spectroscopy (EELS) analysis (Fig. S6) were performed on a double spherical aberration-corrected scanning transmission electron microscope (AC-STEM) in cross-sectional view. The analysis confirms the enrichment of B elements at the TGBs and the enrichment of Cu elements within the grains." (Line 95-98, Page 4).

“Fig. 1. TEM images of the structures of the "bamboo-like" dual-phase Cu-B nanocomposite film. (A) The three-dimensional, reconstructed TEM image. The surface of the sample lies in the x-y plane, and the z axis indicates the depth from the surface. (B) Representative plan-view HRTEM image, which shows the structure of the amorphous region surrounding the crystalline region, and the distance between the crystal planes in the crystalline region is 0.21 nm. (C and D) The Fast Fourier transform (FFT) image of the crystalline region and the amorphous region, respectively. Diffraction spots on the Cu (111) and Cu (200) planes can be distinguished in (C). (E) Representative cross-sectional view HRTEM. Two regions were selected, respectively in the grain interior and near the GB of the columnar crystal structure. (F and G) The inverse Fast Fourier transform (IFFT) image of the two selected regions in (E), respectively. The upper right panel shows the corresponding FFT and filtered (111) diffraction spots. The “⊥” symbols represent dislocations. (H) Plan-view HAADF-STEM image. (I) The element

mapping of Cu corresponding to (H). (J) Cross-sectional view of the HAADF-STEM image. The red arrow shows the position and direction of the EDS line scan. (K) The contrast of the HAADF image (black), Cu content (orange) and B content (blue) as a function of position.”

We have added the following content in the revised Supplementary Information.

“Fig. S6. EELS for the “bamboo-like” dual-phase Cu-B nanocomposite film in cross-sectional view. (A) The EELS imaging reveals discernible columnar arrangements within the film. (B, C and D) Scanned image, low-loss image and high-loss image, respectively, of the selected locations in A. Point scanning is performed at position #1 within the TGB and at position #2 within the grain. (E) High-loss partial spectra at positions #1 and #2, where a more significant peak at around 200 eV is clearly observed at position #1 compared to position #2, which can be identified as the peak of element B⁸. (F) A magnified image of the energy loss around 200 eV, showing that the peak area at position #1 is ~6.2 times larger than that at position #2 after subtracting the background. This result demonstrates the enrichment of element B at the TGB.”

“8. Lu Y-G, Turner S, Ekimov EA, Verbeeck J, Van Tendeloo G. Boron-rich inclusions and boron distribution in HPHT polycrystalline superconducting diamond. *Carbon* **86**, 156-162 (2015).”

Comment 2:

2. It is known that residual stresses develop in sputtered films. The authors shall quantify residual stresses in these Cu-B alloy films.
3. The elastic modulus of the Cu-B films should also be provided.
4. Also have the authors looked at Cu-B alloy films with different B composition? A set of Cu-B films with variable composition will help to establish the solid trend showing the influence of B composition and grain size on mechanical behavior of the Cu-B coatings.

Authors' reply: We thank the reviewer for these comments. In response, we synthesized Cu-B thin films with varying B concentrations and characterized their composition, structure, and mechanical properties by XPS (Fig. S1), XRD (Fig. S2), TEM (Figs. S8 and S9), and nanoindentation (Fig. 2), respectively. We have compiled the relevant information in Table S1 and added the following contents in the revised manuscript:

“The addition of element B causes a significant decrease in the grain size, from 18.9 nm for the pure Cu film to 8.1 nm for the Cu-10.3 at.% B film , 9.5 nm for Cu-26.5 at.% B films, and 6.5 nm for Cu-36.1 at.% B films. The grain size of Cu-B films does not show a regular pattern with increasing B concentration and is close to the critical grain size for Hall-Petch effect failure (Table S1).” (Line 124-127, Page 5).

“We have observed the cross-sectional morphology of the Cu-B films with varying B concentrations. The Cu-10.3 at.% B film shows a morphology without significant features and there is a diffuse distribution of amorphous segregated phases (Fig. S8). Conversely, in the Cu-36.1 at.% B film, the fundamental columnar growth morphology endures, yet a discernible discontinuity exists in the columnar structure, and the progression of columnar grain growth becomes impeded by the amorphous B phase (Fig. S9).” (Line 127-132, Page 5).

“That is, the hardness of Cu-B films shows an increasing then decreasing trend with the increase of B concentration, while the trend of their moduli is consistent with the hardness (Table S1).” (Line 144-146, Page 5).

“We analyzed several factors that may affect the hardness of the films, including residual stresses and grain size. Our measured results show that the films have small residual stresses which do not show a regular trend, indicating that the residual stresses have a negligible effect on the hardness. At the same time, Cu-B films with varying B concentrations have similar grain sizes, thus are not expected to have notably different contributions to hardness (Table S1).” (Line 148-152, Page 5).

We have added the following contents in the revised Supplementary Information.

“Fig. S1. XPS spectra of Cu-B films with varying B concentration. The B 1s and Cu 2p orbitals exhibit distinct peaks corresponding to B-B and Cu-Cu bonds, respectively. The concentration of B element in the films is determined by analyzing the peak areas.”

“Fig. S2. XRD patterns of pure Cu, pure B, and Cu-B films. The pure Cu film shows a single fcc (111) peak, while the Cu-B films exhibit an additional fcc (200) peak. The intensity of the peak becomes progressively weaker with increasing B concentration. The pure B film demonstrates an amorphous structure. Grain size is calculated from the Cu(111) peak in the XRD graph by Scherer's formula ⁷: $d = k\lambda/\beta\cos\theta$, where λ is the wavelength of the X-ray, β is the full width at half maximum (FWHM) of the peak, θ is diffraction angle, and k is a constant.”

“Table S1. Experimental parameters (input power of Cu/B target), composition measured by XPS, residual stress, grain size calculated by XRD, hardness and modulus of the pure Cu and Cu-B alloy films with variable B concentrations.”

Cu/B (W)	Composition			Grain size (nm)	Residual stress (GPa)	Modulus (GPa)
	Cu (at.%)	B (at.%)	B (wt.%)			
20/0	100	0	0	18.9	0.75	103.1 ± 1.6
20/150	89.7	10.3	1.9	8.1	-0.82	143.2 ± 1.7
20/250	73.5	26.5	5.8	9.5	-0.72	176.6 ± 4.6
20/350	63.9	36.1	8.8	6.5	-0.07	156.9 ± 3.3

The hardness of Cu-B films with varying B concentrations is shown in Fig. 2A:

“Fig. 2. Nanoindentation results on the “bamboo-like” dual-phase Cu-B nanocomposite film. (A) Hardness variation with boron concentration for all films synthesized. The gray dashed line is the predicted value based on the rule of mixing (ROM) for pure Cu and pure B. (B) Hardness variation with grain size (d) in logarithmic scale

for the “bamboo-like” dual-phase Cu-B nanocomposite film compared with other binary Cu alloys. The black dashed line indicates the hardness of nanostructured Cu predicted by the Hall-Petch effect. The red cross marks the hardness obtained via a comprehensive evaluation of the boundary strengthening effect (see Supplementary Text). (C) Hardness variation with the solute element weight concentration for the “bamboo-like” dual-phase Cu-B nanocomposite film compared with other binary Cu alloys, and the hardness values of some Cu matrix composites is listed on the left for comparison. The same legends in (B) and (C) are shown in the bottom panel.”

In addition, we provide the cross-sectional TEM images of Cu-10.3 at.% B and Cu-36.1 at.% B films in the revised Supplementary Information.

“Fig. S8. TEM analysis of the Cu-10.3 at.% B film's cross-sectional structures. (A) The representative TEM image shows a morphology without significant features. **(B)** The SAED image shows a standard fcc polycrystalline diffraction ring. **(C to F)** The BF-STEM, DF-STEM, and HAADF-STEM images, along with Cu element mapping (shown at the same position), respectively, corresponding to the blue boxes in (A). These images reveal a featureless morphology with uniform distribution of Cu elements. **(G)** The representative HRTEM image shows slight amorphous phase distributed diffusely between Cu grains. **(H and I)** The HRTEM images of the crystalline and amorphous regions, respectively. The FFT images of the corresponding regions are shown in the upper right corner. The crystalline region shows a complete lattice stripe with FFT results containing Cu(111) and

Cu(200) diffraction spots, indicating the standard fcc structure. The amorphous region shows disordered structure.”

“**Fig. S9. TEM analysis of the Cu-36.1 at.% B film's cross-sectional structures.** (A) The representative TEM image shows that the film exhibits a short columnar structure. (B and C) The representative HAADF-STEM and DF-STEM images show the nanograins embedded in the amorphous phase. (D and E) The HAADF-STEM image and the corresponding elemental mapping of Cu illustrate that the Cu element is mainly distributed inside the grains. (F) The representative HRTEM image presents the morphology of the single grain and surrounding area. (G and H) The HRTEM images of the crystalline and the amorphous regions, respectively. The lattice stripes of both Cu(111) and Cu(200) orientations are present inside the crystalline region. The FFT image of the amorphous region is shown in the upper right corner, presenting a distinct amorphous halo.”

The details of our additional experiments are shown in the **Methods** section of the revised manuscript.

Reviewer #2: The manuscript by Lv et al, “Nanostructured copper with amorphous boron decorated boundaries exhibits record-high hardness,” suggests that the unique bamboo structure created by co-deposition using magnetron sputtering underlies the high hardness of this copper alloy. The work not only reports on the hardness, but also includes detailed STEM observation of the deformed material to relate the deformation mechanism to the structure. The work also includes ab initio molecular dynamics simulation to examine how the deposition process of Cu-B leads to the bamboo structure. Lastly, the authors perform DFT calculation for the solubility of B in Cu. In total, this is a very nice paper.

Authors’ reply: We appreciate the reviewer’s positive appraisal of our work.

Comment 1:

I do not believe, however, that the authors have shown that underlying cause for the high hardness in their Cu-B material is fundamentally different from the nano crystalline-core/amorphous-shell structure explanation (i.e.-supra-nanometer-sized dual-phase glass-crystal) provided in their ref [14] for a Mg based alloy. Possibly the authors could have better related hardness to structure by varying the B concentration, or by showing that a Cu/B nanolaminate structure with comparable layer thicknesses and compositions has significantly lower hardness.

Authors’ reply: We would like to express our gratitude to the reviewer for this valuable suggestion. We synthesized Cu-B films with varying B concentrations of 10.3 at.% and 36.1 at.% and a multilayer film. The multilayer film is comprised of a crystalline Cu layer with a thickness of 10.6 nm and an amorphous B layer with a thickness of 3.6 nm, which has a similar elemental composition and structural dimensions to the “bamboo-like” Cu-B film. We have presented in Fig. 2A the results on the hardness of the films prepared as suggested by the reviewer along with the results of the “bamboo-like” Cu-B film, and the comparison clearly shows the hardness advantage of the “bamboo-like” Cu-B film. The structures of the above films are described in the revised manuscript:

“The Cu-10.3 at.% B film shows a morphology without significant features and there is a diffuse distribution of amorphous segregated phases (Fig. S8). Conversely, in the Cu-36.1 at.% B film, the fundamental columnar growth morphology endures, yet a discernible discontinuity exists in the columnar structure, and the progression of columnar grain growth becomes impeded by the amorphous B phase (Fig. S9). The above results indicate that only a suitable B concentration is conducive to the formation of the “bamboo-like” dual-phase nanocomposite structure, forming continuous and uniform amorphous B phases and maintaining the growth of columnar Cu grains without interruption.” (Line 128-134, Page 5).

“The multilayer is comprised of a crystalline Cu layer with a thickness of ~10.6 nm and an amorphous B layer with a thickness of ~3.6 nm. The Cu layer exhibited a polycrystalline structure with discernible GBs (Fig. S10).” (Line 137-139, Page 5).

“Fig. 2. Nanoindentation results on the “bamboo-like” dual-phase Cu-B nanocomposite film. (A) Hardness variation with boron concentration for all films synthesized. The gray dashed line is the predicted value based on the rule of mixing (ROM) for pure Cu and pure B. (B) Hardness variation with grain size (d) in logarithmic scale for the “bamboo-like” dual-phase Cu-B nanocomposite film compared with other binary Cu alloys. The black

dashed line indicates the hardness of nanostructured Cu predicted by the Hall-Petch effect. The red cross marks the hardness obtained via a comprehensive evaluation of the boundary strengthening effect (see Supplementary Text). (C) Hardness variation with the solute element weight concentration for the “bamboo-like” dual-phase Cu-B nanocomposite film compared with other binary Cu alloys, and the hardness values of some Cu matrix composites is listed on the left for comparison. The same legends in (B) and (C) are shown in the bottom panel.”

We have added the following contents in the revised supplementary information:

“Fig. S8. TEM analysis of the Cu-10.3 at.% B film's cross-sectional structures. (A) The representative TEM image shows a morphology without significant features. **(B)** The SAED image shows a standard fcc polycrystalline diffraction ring. **(C to F)** The BF-STEM, DF-STEM, and HAADF-STEM images, along with Cu element mapping (shown at the same position), respectively, corresponding to the blue boxes in (A). These images reveal a featureless morphology with uniform distribution of Cu elements. **(G)** The representative HRTEM image shows a small amount of amorphous phase distributed diffusely between Cu grains. **(H and I)** The HRTEM images of the crystalline and amorphous regions, respectively. The FFT images of the corresponding regions are shown in the upper right corner. The crystalline region shows a complete lattice stripe with FFT results containing Cu(111) and Cu(200) diffraction spots, indicating the standard fcc structure. The amorphous region shows disordered structure.”

“**Fig. S9. TEM analysis of the Cu-36.1 at.% B film's cross-sectional structures.** (A) The representative TEM image shows that the film exhibits a short columnar structure. (B and C) The representative HAADF-STEM and DF-STEM images show the nanograins embedded in the amorphous phase. (D and E) The HAADF-STEM image and the corresponding elemental mapping of Cu illustrate that the Cu element is mainly distributed inside the grains. (F) The representative HRTEM image presents the morphology of the single grain and surrounding area. (G and H) The HRTEM images of the crystalline and the amorphous regions, respectively. The lattice stripes of both Cu(111) and Cu(200) orientations are present inside the crystalline region. The FFT image of the amorphous region is shown in the upper right corner, presenting a distinct amorphous halo.”

“Fig. S10. TEM analysis of the Cu/B multilayer film's cross-sectional structures. (A) The representative TEM image shows a distinct layer structure. **(B to E)** The BF, DF-STEM, HAADF-STEM images, and the element mapping of Cu at the same position, respectively. The thickness of Cu layer and B layer is 10.6 nm and 3.6 nm, respectively. The elemental composition and structural dimensions are similar to those of the "bamboo-like" Cu-B film. **(F)** The representative HRTEM image containing both Cu layer and B layer. **(G)** HRTEM image of the B layer shows the amorphous structure. **(H)** The FFT image of (G) exhibits an amorphous halo. **(I)** HRTEM image of the Cu layer shows significant GBs. **(J)** The FFT image of (I) displays two sets of Cu (111) diffraction spots, indicating the presence of GBs.”

Comment 2:

Lastly, the authors only provide information about the hardness, but very little regarding other mechanical properties, thermal stability, or potential applications.

Authors' reply: We thank the reviewer for this suggestion. To provide the information on strength and ductility, we performed micropillar compression tests on the films (Fig. 6). The results show a yield strength of ~1.36 GPa and a flow stress of ~2.58 GPa, as well as a failure strain of over 50%. The thermal stability of the films is evaluated by measuring the structure and hardness of films after vacuum annealing at 200 °C for 1h (Fig. S11).

We have added the following contents to the revised manuscript:

“Based on the previous analysis, it is speculated that the potential to restrict the shear behavior of the “bamboo-like” dual-phase nanocomposite structure may impede material failure arising from shear deformation, thereby making a noteworthy contribution towards enhancing ductility. To verify this scenario, we conducted in-situ compression tests on the film, as displayed in the Supplementary Movie. The results indicate that the engineering stress-strain curve during the testing process is remarkably smooth and does not exhibit any pop-in points (Fig. 6A). On the premise that the deformation of the micropillar is uniform, the true stress-strain curve is derived and show a yield strength ($\sigma_{0.2\%}$) of ~ 1.64 GPa and a flow stress (σ_{\max}) of ~ 2.45 GPa (Fig. 6B). However, the deformation process of the micropillar is non-uniform, primarily dominated by barrel-shaped deformation at the top. Hence, by refitting the true stress-strain curve using the real-time measurement of the micropillar's cross-sectional area (Fig. 6C), we obtained a yield strength of ~ 1.36 GPa and a flow stress of ~ 2.58 GPa. Notably, the micropillar exhibited no shear bands or cracks even at strains exceeding 50%, indicating high plasticity and ductility. In contrast, Cu-based micropillars reported in previous studies with similar strength did not exhibit such high ductility^{56,57,58,59}. The results confirm our hypothesis that the “bamboo-like” dual-phase nanocomposite structure constrains the shear behavior, resulting in high strength and ductility. It signifies that the “bamboo-like” dual-phase Cu-B nanocomposite film is substantially hardened while retaining the intrinsic ductility of the metal, thereby achieving remarkable strengthening and toughening.” (Line 280-295 Page 9-10).

“...we confirm that the “bamboo-like” dual-phase Cu-B nanocomposite films have good thermal stability, maintaining the columnar growth morphology and ~ 7.1 GPa hardness (2.7 times that of the annealed pure Cu films) after vacuum annealing at 200 °C for 1 h, which bodes well for their potential application as structural materials in higher temperature environments (Fig. S11).” (Line 162-166 Page 6).

“56. Mara NA, Bhattacharyya D, Dickerson P, Hoagland RG, Misra A. Deformability of ultrahigh strength 5nm Cu/Nb nanolayered composites. *Appl Phys Lett* **92**, (2008).

57. Fan Z, Li Q, Li J, Xue S, Wang H, Zhang X. Tailoring plasticity of metallic glasses via interfaces in Cu/amorphous Cu/Nb laminates. *J Mater Res* **32**, 2680-2689 (2017).

58. Zhang JY, Liu G, Sun J. Self-toughening crystalline Cu/amorphous Cu–Zr nanolaminates: Deformation-induced devitrification. *Acta Mater* **66**, 22-31 (2014).

59. Raghavan R, *et al.* Comparing small scale plasticity of copper-chromium nanolayered and alloyed thin films at elevated temperatures. *Acta Mater* **93**, 175-186 (2015).” (Line 514-521, Page 17).

“Fig. 6. Results of in-situ compression tests of the “bamboo-like” dual-phase Cu-B nanocomposite film. (A) Engineering stress-strain curve. The red dashed line is a linear fit to the elastic regime, where the micropillar shows elastic behavior until the first yield point (σ_0) at a strain ~ 0.06 (ϵ_0). The yield strength $\sigma_{0.2\%} = 1.58$ GPa corresponds to the stress at a strain of $\epsilon_0 + 0.2\%$. (B) True stress-strain curve obtained assuming uniform micropillar deformation. The yield strength $\sigma_{0.2\%} = 1.64$ GPa, while the flow stress $\sigma_{\max} = 2.45$ GPa corresponds to the stress at a strain of $\epsilon_0 + 8\%$. (C) True stress-strain curve obtained by fitting the real-time measurement of the micropillar cross-sectional area. The yield strength $\sigma_{0.2\%} = 1.36$ GPa, while the flow stress $\sigma_{\max} = 2.58$ GPa. Furthermore, the SEM images of the micropillar at different strains are also provided in the figure.”

We have added the following contents in the revised Supplementary Information:

“Fig. S11. Variation of structure and hardness of pure Cu film and “bamboo-like” dual-phase Cu-B nanocomposite film after vacuum annealing at 200 °C for 1 h. (A) XRD patterns of pure Cu film and “bamboo-like” dual-phase Cu-B nanocomposite film before and after annealing. The crystalline quality of pure Cu film is significantly enhanced after annealing and an additional (200) peak appears; the “bamboo-like” dual-phase Cu-B nanocomposite films show a decrease in crystalline quality after annealing and a significant amorphization trend. (B) Grain sizes of pure Cu and “bamboo-like” dual-phase Cu-B nanocomposite film before and after annealing. The grain size of pure Cu film increases from 18.9 nm to 37.4 nm after annealing, while the grain size of “bamboo-like” dual-phase Cu-B nanocomposite film remains almost constant at ~10 nm after annealing. (C to F) Cross-sectional SEM images of pure Cu and “bamboo-like” dual-phase Cu-B nanocomposite films before and after annealing. Before annealing, both films have a columnar growth morphology, whereas the morphology of the pure Cu film changes significantly after annealing, while the “bamboo-like” dual-phase Cu-B nanocomposite film maintains the columnar growth morphology, indicating structural stability. (G) Hardness of pure Cu and “bamboo-like” dual-phase Cu-B nanocomposite films before and after annealing. The “bamboo-like” dual-phase Cu-B nanocomposite film maintains a relatively high hardness of ~7.1 GPa after annealing.”

The details of our additional experiments are shown in the **Methods** section of the revised manuscript.

Reviewer #3:

Comment 1:

The authors present a manuscript on the hardness / strength and deformation mechanisms of a copper - boron nano composite. Beside nano indentation extensive use of electron microscopy is made to identify the microstructure and changes due to plastic deformation by indentation. They found a very high hardness 10.8 GPa and explained this by 3 main processes: (i) indentation induces grain refinement, (ii) strong support of the boron network and (iii) enhanced stress response of the columnar copper grains due to the constrain of the boron network. In the manuscript the authors mention often their material as copper alloy. In my opinion this is misleading as the structure of the current material is a nano composite consisting of columnar copper grains with a diameter of about 11 nm and an amorphous boron network at the "grain boundaries" with a thickness of about 2.5 nm. Hence a comparison with copper alloys is not useful. Overall, the structure and also the deformation processes looks similar to that of nanoscale multilayers. In fact, I am missing here some literature comparison of hardness and deformation patterning (e.g. bending of the layers).

Authors' reply: We appreciate the valuable comments by the reviewer. As the reviewer pointed out, the "bamboo-like" dual-phase Cu-B nanocomposite film we synthesized differs significantly from conventional alloy materials. Therefore, we have deleted the expression "Cu-B alloy films" in the revised manuscript and changed it to "Cu-B films". We understand that alloying and constructing composite systems are both highly effective methods for enhancing the strength of metallic materials. Therefore, we compared the hardness of our synthesized film with binary Cu alloy films and various Cu matrix composites, including B-Cu composite, Cu-30% SiC composite, Cu-50% SiC composite, Cu-MoS₂-WC composite, amorphous C/Cu composite, graphene/Cu composite and Cu-carbon nanotube composite. The results show that the hardness of "bamboo-like" dual-phase Cu-B nanocomposite film is the highest among them.

In light of the similarity of the structure and deformation process to nanoscale multilayers that the reviewer mentioned, we synthesized a Cu/B multilayer film. The multilayer film is comprised of a crystalline Cu layer with a thickness of 10.6 nm and an amorphous B layer with a thickness of 3.6 nm, which has a similar elemental composition and structural dimensions to the "bamboo-like" Cu-B film (Fig. S10). Subsequently, we performed a nanoindentation test and observed the resulting structure beneath the indentation (Fig. S14). The results show that the Cu/B multilayer film we synthesized has distinct shear bands after deformation, which is consistent with the behaviors of the multilayers reported in the literature that exhibit layer bending phenomena. The deformation process is fundamentally different from that of the "bamboo-like" dual-phase Cu-B nanocomposite film presented in the present manuscript.

We have provided relevant descriptions in the revised manuscript:

“...whereas the similar region of the Cu/B multilayer film does not produce any bending phenomenon and only a slight decrease in layer thickness (Figs. S13B and S13C).” (Line 181-182, Page 6).

“In contrast, the multilayer film sprouts a shear band across the film thickness in the region of larger deformation (Figs. S13F and S13G), with obvious faults on both sides of the shear band (Figs. S13H and S13I) but without the phenomenon of layer bending. It should be noted that nanoscale multilayers with layer bending during deformation are usually accompanied by shear bands^{45,46,47,48,49}, and the deformation mechanism is significantly different from the column structure bending of the "bamboo-like" dual-phase Cu-B nanocomposite film.” (Line 189-194 Page 6-7).

“In the region just below the indenter tip, the complex stress conditions produce significant grain refinement (Figs. 3N and 3O), producing a distinct GB morphology. IFFT images clearly distinguish random orientation changes of the (111) lattice fringes (Figs. 3P and 3Q). Similar changes are produced in Cu/B multilayers.” (Line 194-196, Page 7).

“45. Fu K, *et al.* Plastic behaviour of high-strength lightweight Al/Ti multilayered films. *J Mater Sci* **52**, 13956-13965 (2017).

46. Lee C-M, Jeng RJ, Yu C-C, Chang C-H, Li C-L, Chu JP. Mechanical property evaluations of an amorphous metallic/ceramic multilayer and its role in improving fatigue properties of 316L stainless steel. *Mater Sci Eng, A* **671**, 198-202 (2016).

47. Li N, Wang H, Misra A, Wang J. In situ nanoindentation study of plastic co-deformation in Al-TiN nanocomposites. *Sci Rep* **4**, 6633 (2014).

48. Verma N, Jayaram V. The influence of Zr layer thickness on contact deformation and fracture in a ZrN–Zr multilayer coating. *J Mater Sci* **47**, 1621-1630 (2011).

49. Major L, Major R, Kot M, Lackner JM, Major B. Ex situ and in situ nanoscale wear mechanisms characterization of Zr/Zr x N tribological coatings. *Wear* **404-405**, 82-91 (2018).” (Line 490-500, Page 16-17).

We have added the following contents in the revised Supplementary Information:

“Fig. S10. TEM analysis of the Cu/B multilayer film's cross-sectional structures. (A) The representative TEM image shows a distinct layer structure. **(B to E)** The BF, DF-STEM, HAADF-STEM images, and the element mapping of Cu at the same position, respectively. The thickness of Cu layer and B layer is 10.6 nm and 3.6 nm, respectively. The elemental composition and structural dimensions are similar to those of the "bamboo-like" Cu-B film. **(F)** The representative HRTEM image containing both Cu layer and B layer. **(G)** HRTEM image of the B layer shows the amorphous structure. **(H)** The FFT image of (G) exhibits an amorphous halo. **(I)** HRTEM image of the Cu layer shows significant GBs. **(J)** The FFT image of (I) displays two sets of Cu (111) diffraction spots, indicating the presence of GBs.”

“Fig. S14. TEM analysis of the indented Cu/B multilayer film's cross-sectional structures. (A) The representative TEM image. (B and C) HAADF-STEM and DF-STEM images of the region with lower plastic strains. (D and E) HAADF-STEM and DF-STEM images of the region with higher plastic strains. (F and G) HAADF-STEM and DF-STEM images of the region with the largest observed plastic strains. (H) TEM image of the shear band region. (I) Magnified HRTEM image of the yellow box in (H).”

Comment 2:

The description of the deformation processes is a little bit speculative. All the mentioned processes make sense and it is likely that they contribute to the hardness increase, however, their contribution is not quantified or discussed which one is maybe the most important.

Authors' reply: We thank the reviewer for this comment. Quantifying the contribution of the deformation process to the hardness increase is a huge challenge. To clarify the contribution of the three main processes produced by the "bamboo-like" structure to the increase in hardness, we have performed additional experiments.

First, we synthesized Cu-B films with varying, both lower (10.3 at.%) and higher (36.1 at.%) B concentrations, and characterized their structures in detail. These Cu-B films have similar grain sizes; however, neither the higher nor the lower B concentration succeeds in building the "bamboo-like" dual-phase nanocomposite structure, and with lower hardness (Fig. 2) even when the B concentration is higher. After excluding the effect of residual stress on hardness, we find that the “bamboo-like” dual-phase Cu-B nanocomposite film exhibits the largest hardness compared with other Cu-B films. This result indicates that B content is not a main contributor to the hardness increase, implying a direct contribution of "enhanced stress response of the nanocolumnar copper structure

constrained by the TGBs" to the hardness increase.

Furthermore, we have compared the deformation process of the Cu/B multilayer film with similar composition and structural dimensions with that of the "bamboo-like" dual-phase Cu-B nanocomposite film. The results show that, in addition to the "indentation induced grain refinement" which is common to both films, the "strong support of the TGBs consisting of an amorphous boron framework" and the "enhanced stress response of the nanocolumnar copper structure constrained by the TGBs" of the "bamboo-like" dual-phase Cu-B nanocomposite film contribute directly to the hardness increase.

Overall, the record-high hardness is achieved by the synergy of multiple mechanisms during the deformation process, which is caused by the specificity of the microstructure. Among these mechanisms, we believe that the most important one is the "Strengthening by enhanced stress response of constrained nanocolumnar copper".

Please refer to our responses to **Reviewer #1's comment 2** and **Reviewer #2's comment 1** for the pertinent new contents in the revised manuscript.

Also, the revised Fig. 2 has been added to the revised manuscript:

“Fig. 2. Nanoindentation results on the “bamboo-like” dual-phase Cu-B nanocomposite film. (A) Hardness variation with boron concentration for all films synthesized. The gray dashed line is the predicted value based on the rule of mixing (ROM) for pure Cu and pure B. (B) Hardness variation with grain size (d) in logarithmic scale for the “bamboo-like” dual-phase Cu-B nanocomposite film compared with other binary Cu alloys. The black

dashed line indicates the hardness of nanostructured Cu predicted by the Hall-Petch effect. The red cross marks the hardness obtained via a comprehensive evaluation of the boundary strengthening effect (see Supplementary Text). (C) Hardness variation with the solute element weight concentration for the “bamboo-like” dual-phase Cu-B nanocomposite film compared with other binary Cu alloys, and the hardness values of some Cu matrix composites is listed on the left for comparison. The same legends in (B) and (C) are shown in the bottom panel.”

Comment 3:

A quick literature review (e.g. Atomic arrangement and mechanical properties of chemical-vapor-deposited amorphous boron, Jessica M. Maita, Gyuho Song Mariel, Colby Seok-Woo Lee, *Materials & Design*, Vol. 193, August 2020 or Amorphous boron coatings produced with vacuum arc deposition technology, C. C. Klepper, R. C. Hazelton, E. J. Yadlowsky, E. P. Carlson, and M. D. Keitz, *Journal of Vacuum Science & Technology A* 20, 725 (2002)) shows that the nano hardness of amorphous boron thin films is between 30 and 35 GPa. If one assumes an area fraction of the amorphous boron of about 25 to 30% (see Fig. 1A top view where the indentation is performed or Fig. 1H) the measured hardness with 10.8 GPa can be explained by a simple parallel composite model. Hence, all the other discussed mechanisms are not necessary. A more in-depth discussion is needed here to justify the proposed mechanisms and what contribution to hardness they really have.

Authors’ reply: We appreciate the reviewer’s suggestion, which inspired us to discuss the contribution of B concentration to the mechanical properties.

For this purpose, we synthesized a pure B film using the same B target under identical experimental parameters as used for the “bamboo-like” dual-phase Cu-B nanocomposite film, resulting in a measured hardness of approximately 19.6 GPa. We performed a rule of mixing (ROM) calculation in combination with the hardness of the pure Cu film, which predicted a much lower value than the "bamboo-like" dual-phase Cu-B nanocomposite film (Fig. 2).

Both Cu-B films with varying B concentrations and Cu/B multilayers with similar composition and structural dimensions exhibit lower hardness. The contribution of the proposed deformation mechanisms to hardness can be demonstrated by comparing them with their structures and deformation processes, as discussed in the responses to the previous comments.

We have added the following content in the revised manuscript:

“We performed nanoindentation tests on the synthesized films mentioned above in the continuous stiffness measurement (CSM) mode in order to determine their hardness (Fig. 2A). The results indicate that the "bamboo-like" dual-phase Cu-B nanocomposite film exhibits the largest hardness of 10.8 ± 0.3 GPa compared with other Cu-B films and exceeds the hardness of the Cu/B multilayer film. That is, the hardness of Cu-B films shows an

increasing then decreasing trend with the increase of B concentration, while the trend of their moduli is consistent with the hardness (Table S1). Furthermore, the hardness of the Cu-B system is predicted using the rule of mixing (ROM), taking into account the hardness values of pure Cu and pure B films, while the "bamboo-like" dual-phase Cu-B nanocomposite film demonstrates considerably higher hardness than this prediction.” (Line 141-148, Page 5)

We have added the following contents in the revised Supplementary Information:

“Fig. S8. TEM analysis of the Cu-10.3 at.% B film's cross-sectional structures. (A) The representative TEM image shows a morphology without significant features. (B) The SAED image shows a standard fcc polycrystalline diffraction ring. (C to F) The BF-STEM, DF-STEM, and HAADF-STEM images, along with Cu element mapping (shown at the same position), respectively, corresponding to the blue boxes in (A). These images reveal a featureless morphology with uniform distribution of Cu elements. (G) The representative HRTEM image shows slight amorphous phase distributed diffusely between Cu grains. (H and I) The HRTEM images of the crystalline and amorphous regions, respectively. The FFT images of the corresponding regions are shown in the upper right corner. The crystalline region shows a complete lattice stripe with FFT results containing Cu(111) and Cu(200) diffraction spots, indicating the standard fcc structure. The amorphous region shows disordered structure.”

“Fig. S9. TEM analysis of the Cu-36.1 at.% B film's cross-sectional structures. (A) The representative TEM image shows that the film exhibits a short columnar structure. **(B and C)** The representative HAADF-STEM and DF-STEM images show the nanograins embedded in the amorphous phase. **(D and E)** The HAADF-STEM image and the corresponding elemental mapping of Cu illustrate that the Cu element is mainly distributed inside the grains. **(F)** The representative HRTEM image presents the morphology of the single grain and surrounding area. **(G and H)** The HRTEM images of the crystalline and the amorphous regions, respectively. The lattice stripes of both Cu(111) and Cu(200) orientations are present inside the crystalline region. The FFT image of the amorphous region is shown in the upper right corner, presenting a distinct amorphous halo.”

Comment 4:

Also tensile properties (ductility) and fracture toughness would be of interest (I know this was not the scope of the current work, however, for a "good" material these parameters are also essential).

Authors' reply: This is a good suggestion. The hardness-ductility trade-off is a crucial issue that has long been faced by the field of structural materials, and it is a fundamental measurement of a “good” material that cannot be overlooked. To obtain a more comprehensive understanding of the mechanical properties of the films, we strived to conduct further tests on them. However, it is challenging to obtain tensile properties for film materials.

Conversely, the compression test provides a more widely applicable method for mechanical properties testing of thin film materials, facilitating a comparison with the existing literature. Therefore, we performed micropillar compression tests on the films to determine their strength and ductility. The results reveal a yield strength of ~1.36 GPa and a flow stress of ~2.58 GPa, as well as a failure strain of over 50%. These findings provide valuable insights into the overall mechanical behavior of the films.

The results of the micropillar compression tests are added to the revised manuscript as follows:

“Based on the previous analysis, it is speculated that the potential to restrict the shear behavior of the “bamboo-like” dual-phase nanocomposite structure may impede material failure arising from shear deformation, thereby making a noteworthy contribution towards enhancing ductility. To verify this scenario, we conducted in-situ compression tests on the film, as displayed in the Supplementary Movie. The results indicate that the engineering stress-strain curve during the testing process is remarkably smooth and does not exhibit any pop-in points (Fig. 6A). On this premise that the deformation of the micropillar is uniform, the true stress-strain curve is derived and show a yield strength ($\sigma_{0.2\%}$) of ~1.64 GPa and a flow stress (σ_{\max}) of ~2.45 GPa (Fig. 6B). However, the deformation process of the micropillar is non-uniform, primarily dominated by barrel-shaped deformation at the top. Hence, by refitting the true stress-strain curve using the real-time measurement of the micropillar's cross-sectional area (Fig. 6C), we obtained a yield strength of ~1.36 GPa and a flow stress of ~2.58 GPa. Notably, the micropillar exhibited no shear bands or cracks even at a strain exceeding 50%, indicating high plasticity and ductility. In contrast, Cu-based micropillars reported in previous studies with similar strength did not exhibit such high ductility^{56,57,58,59}. The results confirm our hypothesis that the "bamboo-like" dual-phase nanocomposite structure constrains the shear behavior, resulting in high strength and ductility. It signifies that the “bamboo-like” dual-phase Cu-B nanocomposite film is substantially hardened while retaining the intrinsic ductility of the metal, thereby achieving remarkable strengthening and toughening.” (Line 280-295 Page 9-10).

“56. Mara NA, Bhattacharyya D, Dickerson P, Hoagland RG, Misra A. Deformability of ultrahigh strength 5nm Cu/Nb nanolayered composites. *Appl Phys Lett* **92**, (2008).

57. Fan Z, Li Q, Li J, Xue S, Wang H, Zhang X. Tailoring plasticity of metallic glasses via interfaces in Cu/amorphous CuNb laminates. *J Mater Res* **32**, 2680-2689 (2017).

58. Zhang JY, Liu G, Sun J. Self-toughening crystalline Cu/amorphous Cu–Zr nanolaminates: Deformation-induced devitrification. *Acta Mater* **66**, 22-31 (2014).

59. Raghavan R, *et al.* Comparing small scale plasticity of copper-chromium nanolayered and alloyed thin films at elevated temperatures. *Acta Mater* **93**, 175-186 (2015).” (Line 514-521, Page 17).

“Fig. 6. Results of in-situ compression tests of the “bamboo-like” dual-phase Cu-B nanocomposite film. (A) Engineering stress-strain curve. The red dashed line is a linear fit to the elastic regime, where the micropillar shows elastic behavior until the first yield point (σ_0) at a strain ~ 0.06 (ϵ_0). The yield strength $\sigma_{0.2\%} = 1.58$ GPa corresponds to the stress at a strain of $\epsilon_0 + 0.2\%$. (B) True stress-strain curve obtained assuming uniform micropillar deformation. The yield strength $\sigma_{0.2\%} = 1.64$ GPa, while the flow stress $\sigma_{\max} = 2.45$ GPa corresponds to the stress at a strain of $\epsilon_0 + 8\%$. (C) True stress-strain curve obtained by fitting the real-time measurement of the micropillar cross-sectional area. The yield strength $\sigma_{0.2\%} = 1.36$ GPa, while the flow stress $\sigma_{\max} = 2.58$ GPa. Furthermore, the SEM images of the micropillar at different strains are also provided in the figure.”

Comment 5:

Overall, the manuscript presents some interesting work on the production of a Cu-B nano composite which can maybe extended to other material combinations. On the other side, the explanations of the so-called "record-high" hardness are insufficient. In my opinion, the hardness is not really "record high" and can be explained by the high hardness of the amorphous boron network by a simple composite model. Also, I am missing the interesting comparison with nanostructured multilayer thin films because some of the mentioned deformation mechanisms are also applying there.

I think the manuscript doesn't meet the high standard of Nature Communication and I don't recommend

publication.

Authors' reply: We appreciate the reviewer's thorough reading of the manuscript and insightful comments. To confirm the record-high hardness of the "bamboo-like" dual-phase Cu-B nanocomposite film, we compared its hardness with that of binary alloy films and various Cu-based composites. The results show that the "bamboo-like" duplex Cu-B nanocomposite film has the highest hardness. In addition, the hardness value is much higher than that calculated by the rule of mixing (ROM) model. We also added results for multilayer films with similar elemental composition and structure size to the "bamboo-like" Cu-B films, but they have different deformation mechanisms. Following the comments and suggestions from this and other reviewers, we have made extensive and substantive revisions in the manuscript to enhance the quality of the content and improve the presentation and discussion. We feel confident that we have adequately addressed the reviewer's concerns, and we eagerly anticipate receiving positive assessments on our considerably amended and improved work.

REVIEWERS' COMMENTS

Reviewer #1 (Remarks to the Author):

The authors have made substantial efforts to address the reviewer's comments. The revised manuscript is much stronger and acceptable for publication.

Reviewer #2 (Remarks to the Author):

The authors have addressed the several issues raised by the three referees, providing much additional information to the manuscript. While the manuscript is now suitable for publication in Nature Communications, a few revisions seem in order.

The key to this work is the growth of a composite (Cu/B) columnar grain structure with a 10 nm lateral length scale, referred to as bamboo-like structure. There have been several experimental studies attempting to use magnetron sputtering to create lateral concentration fluctuations in metal films as well as models developed to gain an understanding of the growth mechanisms. For example:

- M. Atzmon, D. Kessler D. Srolovitz, J. Appl. Phys. 72, 442 (1992).
- C. Adams, D. Srolovitz, and M. Atzmon, J. Appl. Phys. 74, 1707 (1993).
- Raghavan, R., Mukherjee, A., & Ankit, K. (2020). Journal of Applied Physics, 128(17), 175303.
- Stewart, J. A., & Dingreville, R. (2020). Acta Materialia, 188, 181-191.
- Raeker, E., Powers, M., & Misra, A. (2020). Scientific Reports, 10(1), 17775

While the present study is not concerned with the growth process of the film, it would be very helpful for the broader aspects of this work if the authors provided context of their work in terms of these past studies. For example, are the authors aware of other systems where similar structures have been grown, or is Cu-B somewhat unique? The authors' present comparison of their work to systems undergoing GB segregation and GB strengthening seems misleading. For example, their ref. [30] concerns quasi-equilibrium segregation and does not seem relevant to the apparent self-organized growth observed here.

The authors might also consider revising or simply removing their discussion of the temperature stability of the Cu-B films. The conclusion that the films are stable during annealing to 200 C and that they therefore useful for high temperature applications is not convincing; the hardness drops ~ 30 %, becoming nearly the same value as the multilayer film Cu-B film.

Reviewer #3 (Remarks to the Author):

The authors have significantly improved the manuscript and addressed my questions/comments

successfully. They also added new results for Cu/B multilayers and compared the resulting hardness with the bamboo Cu/B structure. They find that the hardness of the bamboo structure is significantly higher than the Cu/B film structure. However, one have to account for the different loading condition between film and bamboo structure here: in the films the indentation was performed perpendicular to the Cu/B layers and in the bamboo structure parallel to the "layers". Would be interesting what the impact on mechanical properties this have.

Anyway the manuscript is now ready for publication.

Responses to the comments by reviewers

Reviewer #1: The authors have made substantial efforts to address the reviewer's comments. The revised manuscript is much stronger and acceptable for publication.

Authors' reply: We appreciate the reviewer's positive appraisal of our work.

Reviewer #2: The authors have addressed the several issues raised by the three referees, providing much additional information to the manuscript. While the manuscript is now suitable for publication in Nature Communications, a few revisions seem in order.

Authors' reply: We appreciate the reviewer's positive appraisal of our work.

Comment 1:

The key to this work is the growth of a composite (Cu/B) columnar grain structure with a 10 nm lateral length scale, referred to as bamboo-like structure. There have been several experimental studies attempting to use magnetron sputtering to create lateral concentration fluctuations in metal films as well as models developed to gain an understanding of the growth mechanisms. For example:

- M. Atzmon, D. Kessler D. Srolovitz, J. Appl. Phys. 72, 442 (1992).
- C. Adams, D. Srolovitz, and M. Atzmon, J. Appl. Phys. 74, 1707 (1993).
- Raghavan, R., Mukherjee, A., & Ankit, K. (2020). Journal of Applied Physics, 128(17), 175303.
- Stewart, J. A., & Dingreville, R. (2020). Acta Materialia, 188, 181-191.
- Raeker, E., Powers, M., & Misra, A. (2020). Scientific Reports, 10(1), 17775

While the present study is not concerned with the growth process of the film, it would be very helpful for the broader aspects of this work if the authors provided context of their work in terms of these past studies. For example, are the authors aware of other systems where similar structures have been grown, or is Cu-B somewhat unique? The authors' present comparison of their work to systems undergoing GB segregation and GB strengthening seems misleading. For example, their ref. [30] concerns quasi-equilibrium segregation and does not seem relevant to the apparent self-organized growth observed here.

Authors' reply: We thank the reviewer for raising this issue. Our manuscript provides a description of the distinctive characteristics exhibited by the Cu-B system, while the insolubility of the Cu-B system has been validated through formation energy calculations and molecular dynamics simulations. These findings have a

significant impact on the growth mechanism of the films. Thus far, no comparable structural and mechanical responses have been identified in the other systems.

The thick grain boundaries (TGBs) constructed by annealing and the grain boundary segregation effect are described in ref. [30]. Although the formation process of TGBs differs from that observed in the "bamboo-like" dual-phase Cu-B nanocomposite film, there exists a structural consistency that justifies a comparative analysis in terms of both structure and strengthening mechanisms.

We have added the following to the revised manuscript: “Previous theoretical and phase-field simulation work investigated the effects of diffusion kinetics and deposition parameters on the phase separation and nanostructure of binary thin films.^{31, 32, 33, 34}” (Line 110-111, Page 4).

“31. Atzmon M, Kessler DA, Srolovitz DJ. Phase separation during film growth. *J Appl Phys* **72**, 442-446 (1992).

32. Adams CD, Srolovitz DJ, Atzmon M. Monte Carlo simulation of phase separation during thin - film codeposition. *J Appl Phys* **74**, 1707-1715 (1993).

33. Stewart JA, Dingreville R. Microstructure morphology and concentration modulation of nanocomposite thin-films during simulated physical vapor deposition. *Acta Mater* **188**, 181-191 (2020).

34. Raghavan R, Mukherjee A, Ankit K. Nanostructural evolution in vapor deposited phase-separating binary alloy films of non-equimolar compositions: Insights from a 3D phase-field approach. *J Appl Phys* **128**, (2020).”

(Line 471-479, Page 15).

Comment 2:

The authors might also consider revising or simply removing their discussion of the temperature stability of the Cu-B films. The conclusion that the films are stable during annealing to 200 C and that they therefore useful for high temperature applications is not convincing; the hardness drops ~ 30 %, becoming nearly the same value as the multilayer film Cu-B film.

Authors' reply: We thank the reviewer for the suggestion, and we have removed the discussion on thermal stability.

Reviewer #3: The authors have significantly improved the manuscript and addressed my questions/comments successfully. They also added new results for Cu/B multilayers and compared the resulting hardness with the bamboo Cu/B structure. They find that the hardness of the bamboo structure is significantly higher than the Cu/B film structure. However, one have to account for the different loading condition between film and bamboo structure here: in the films the indentation was performed perpendicular to the Cu/B layers and in the bamboo structure parallel to the "layers". Would be interesting what the impact on mechanical properties this have. Anyway the manuscript is now ready for publication.

Authors' reply: We appreciate the valuable comments by the reviewer. In response, we have provided relevant descriptions in the revised manuscript: “It is worth noting that when the loading direction is perpendicular to the thin film, there exists a strong correlation between the structure and the loading. Enhancing the ability to suppress shear can effectively improve strength. Although multilayer structures can enhance strength by directly impeding dislocation motion at layer interfaces, bamboo-like structures can achieve better resistance to shear processes through improved bending response, thus demonstrating superior performance.” (Line 282-286, Pages 9-10).